# Development of level 2 aerosol and surface products from cross-track scanning polarimeter POSP onboard GF-5(02) satellite

Cheng Chen[1], Xuefeng Lei[1], Zhenhai Liu[1], Haorang Gu[2], Oleg Dubovik[3], Pavel Litvinov[4], David Fuertes[4], Yujia Cao[1,5], Haixiao Yu[1,5], Guangfeng Xiang[1], Binghuan Meng[1], Zhenwei Qiu[1], Xiaobing Sun[1], Jin Hong[1], Zhengqiang Li[2]

[1] Anhui Institute of Optics and Fine Mechanics, Hefei Institutes of Physical Science, Chinese Academy of Sciences, Hefei 230031, China
[2] State Environment Protection Key Laboratory of Satellite Remote Sensing, Aerospace Information Research Institute, Chinese Academy of Sciences, Beijing 100101, China
[3] Univ. Lille, CNRS, UMR 8518 - LOA - Laboratoire d'Optique Atmosphérique, F-59000 Lille, France
[4] GRASP-SAS, Satellite Remote Sensing Development, Lille 59260, France
[5] University of Science and Technology of China, Hefei 230036, China

*Correspondence to*: Zhengqiang Li (lizq@radi.ac.cn)

**Abstract.** Development of long-term continuous, consistent and high-quality satellite remote sensing aerosol and surface products is crucial to constrain climate models and improve our understanding on climate change. Particulate Observing Scanning Polarization (POSP) is the first space-borne multi-spectral (UV-VIS-NIR-SWIR) cross-track scanning polarimeter dedicated to complement Directional Polarimetric Camera (DPC) multi-angular polarimetric measurements and perform synergetic observations namely the Polarization Cross-Fire suite (PCF) onboard Chinese Gaofen series GF-5(02) satellite. The POSP unique single-viewing spectral (UV-VIS-NIR-SWIR) high-precision polarimetric measurements provide rich information for atmospheric aerosol and surface characterization. Here we developed aerosol and surface products from POSP/GF-5(02) based on the Generalized Retrieval of Atmosphere and Surface Properties (GRASP)/Models approach. The detailed retrieval approach and processing scheme are provided. The baseline level 2 product was generated for the first 18 months POSP measurements from December 2021 to May 2023 and publicly available and registered at doi.org/10.57760/sciencedb.14748. The obtained POSP/GF-5(02) aerosol and surface products are validated and intercompared with ground-based AERONET reference aerosol dataset and other independent satellite products, such as NOAA-20 VIIRS/DB aerosol product and MODIS MCD43 surface product. The results show generally good consistency of POSP products with AERONET, VIIRS/NOAA-20 aerosol dataset and MODIS surface product. Moreover, the developed POSP product includes not only total Aerosol Optical Depth (AOD), but also detailed properties of aerosol such as aerosol size, absorption, layer height, type, etc., as well as full surface Bidirectional Reflectance Distribution Function (BRDF), Bidirectional Polarization Distribution Function (BPDF), and black-sky, white-sky albedos and Normalized Difference

Vegetation Index (NDVI). These parameters are of high importance to constrain the Earth-atmosphere radiation budget. The retrieval of these properties seems to be possible due to the polarimetric capabilities and wide UV-VIS-NIR-SWIR spectral range of POSP observations and advances of used GRASP/Models retrieval approach. Finally, some potential improvements for POSP level 1 to level 2 processing chain are identified, and limitations and lessons learned are discussed.

## 1 Introduction

Satellite remote sensing provides an important source of data for dealing with the climate change challenges as well as monitoring environmental health. Therefore, the accumulation of high-quality long-term consistent remote sensing observational data has been paid more and more attention. Especially for climate change studies, the establishment of long-term and high-precision satellite remote sensing products, despite of huge challenges, is particularly important for reducing the uncertainty of climate change assessments (IPCC, 2021). Space agencies and even private companies are launching or planning to launch an increasing number of Earth-observing sensors. In particular, polarimetric remote sensing is widely regards as one of the best technologies for detecting atmospheric aerosols and clouds (Dubovik et al., 2019, 2021b; Hansen et al., 1997; Mishchenko et al., 2004). The deployment of the first multi-angular polarimeter (MAP) payload POLDER (Polarization and Directionality of the Earth's Reflectances) series (POLDER-1, -2 and -3) (Bréon et al., 2011; Deschamps et al., 1994; Tanré et al., 2011) developed by French national space agency (CNES), was followed by HARP-2 (Hyper-Angular Rainbow Polarimeter) and SPEXOne (Spectro-Polarimetric Experiment) recently launched on NASA's PACE (Plankton, Aerosol, Cloud, ocean Ecosystem) mission (Hasekamp et al., 2019; Martins et al., 2018; Remer et al., 2019a, b). In addition, the first commercial multi-angular cub-sat polarimeter of GAPMAP series has been launched by GRASP-Earth (Fuertes et al., 2023; Martins et al., 2024). The richness of multi-angular polarimetric is evident and demonstrated in numerous applications (Cairns et al., 2009; Chen et al., 2020, 2022a; Dubovik et al., 2011, 2019; Fu et al., 2020; Fu and Hasekamp, 2018; Gao et al., 2019; Hasekamp et al., 2024, 2011; Knobelspiesse et al., 2020; Waquet et al., 2009; Xu et al., 2016, 2017a, b). At the same, the community also recognizes the challenges in extracting the large number of parameters from polarimetric observations (e.g., Dubovik et al., 2019; Chen et al., 2020). Therefore, in order to leverage the information richness of multi-angular polarimetric measurements on a global scale releasing different levels of products, and sharing the successes and challenges with the broader community is crucial.

China has recently launched several payloads with polarimetric capabilities. Most of these sensors were designed and developed by the Anhui Institute of Optics and Fine Mechanics, Chinese Academy of Sciences (CAS) (Chen et al., 2021b; Li et al., 2018, 2022b, c, d). However, the past efforts were focused on the development of instruments, while development of user-end products, and in-depth product analysis was lacking. For example, the Directional Polarimetric Camera (DPC), which is the first POLDER-like Chinese operational MAP sensor, has been launched on-board GaoFen-5 (GF-5) satellite in 2018 (Li et al., 2018) and several similar instruments are deployed on different platforms afterwards. There several case

studies (regional or short-term) that use the data to determine aerosol and cloud properties have been published (Chen et al., 2021a; Ge et al., 2022; Jin et al., 2022; Lei et al., 2023; Li et al., 2021, 2022a; Shang et al., 2020; Wang et al., 2022; Yu et al., 2024; Zhang et al., 2023), however reliable information about the released global product as well as demonstration of their use for environment and climate applications is very limited.

In this study, we will focus on the development of global aerosol and surface products from a new polarimeter (POSP), the first space-borne UV-VIS-NIR-SWIR multi-spectral cross-track scanning polarimeter, onboard GF-5(02) satellite. POSP was designed to complement DPC multi-angular polarimetric measurements of I, Q, U at VIS-NIR channels conducted on the same platform with maximum 17 viewing angles and ~3.3 km spatial resolution. POSP extends to include UV and SWIR

channels, which is expected to enhance the atmospheric aerosol detection capability, particularly for aerosol layer height, fine/coarse mode, to achieve the main goal of the GF-5(02) mission that is dedicated for PM$_{2.5}$ remote sensing (Li et al., 2022). Meanwhile due to the on-board calibration device for POSP, it was expected to obtain higher accuracy of intensity/polarization measurements than DPC, which could perform cross-calibration between DPC and POSP (Lei et al., 2023). Therefore, the cross-track pattern was chosen to achieve more overlaps. The GRASP/Models approach, which has

previously been successfully applied to multi-angle polarimeter (Chen, et al., 2020, Dubovik et al., 2021b, etc.) and multi-spectral spectrometer (UV-VIS-NIR-SWIR) (Litvinov et al., 2024 and Chen et al., 2024), is adapted to process POSP single-viewing UV-VIS-NIR-SWIR polarimeter measurements. In the previous development of TROPOMI/Sentinel-5p aerosol and surface products, within the framework of ESA S5p+Innovation AOD/BRDF project (Litvinov et al., 2022), we have demonstrated that the possibility for extend aerosol and surface characterization from single-view UV-VIS-NIR-SWIR

measurements with properly constrained forward models and advanced retrieval algorithm (Litvinov et al., 2024; Chen et al., 2024a). We describe the POSP/GF-5(02) Level 1 measurements and global aerosol/surface datasets used in this study in Section 2. Section 3 describes the POSP aerosol and surface retrieval scheme and the Level 2 product specification. Section 4 illustrates the preliminary validation and evaluation of POSP Level 2 aerosol and surface product. The study is concluded in Section 5 with a summary and discussion of the lessons learned.

**2 Data description**

In this section, the POSP/GF-5(02) satellite Level 1 measurements and the independent ground-based and space-borne aerosol and surface datasets used in this study are described.

**2.1 Single-viewing scanning polarimeter POSP/GF-5(02) Level 1 data**

Gaofen (GF) is a series of Chinese civilian remote sensing satellites specifically for China's high resolution Earth

observation system (Chen et al., 2022c; Gu and Tong, 2015; Tong et al., 2016). Among them, GF-5 satellite is designed and dedicated for atmospheric detection and environment monitoring, including aerosol, cloud, water vapor, ozone as well as

tracer gases. The first GF-5(01) satellite was launched on May 8, 2018. GF-5(01) carries 6 payloads, specifically Directional Polarization Camera (DPC), Environment Monitoring Instrument (EMI), Greenhouse-gases Monitoring (GMI), Atmospheric Infrared Ultra-spectral Sensor (AIUS), Visual Infrared Multispectral Sensor (VIMS) and Advanced Hyperspectral Imager

(AHSI). GF-5(01) satellite continued working until March 2021. Subsequently, the second GF-5(02) satellite was launched on September 7, 2021. Generally, GF-5(02) keeps similar design with GF-5(01). In order to enhance the atmospheric detection capability, GF-5(02) deployed a new Particulate Observing Scanning Polarization (POSP) that is a cross-track scanning polarimeter to complement DPC and perform synergetic observations, namely the Polarization Cross-Fire suite (Li et al., 2022d). The original POSP design idea came from NASA Aerosol Polarimetry Sensor (APS) that was an along track

scanning polarimeter and lost during unsuccessful launch of Glory mission in 2011 (Dubovik et al., 2019; Mishchenko et al., 2007). POSP is the first space-borne multi-spectral cross-track scanning polarimeter, covering UV-VIS-NIR-SWIR spectrum (Lei et al., 2020). Specially, an on-board calibration device is designed for POSP to perform on-orbit solar diffuse reflector-based radiometric calibration as well as polarimetric calibration. Hence, it is expected to obtain high-precision polarimetric measurements from POSP achieving cross-calibration between DPC and POSP (Lei et al., 2023). Table 1 summaries the

POSP/GF-5(02) main sensor characteristics and spectral channels used in this study.

**Table 1. Instrument characteristics for POSP onboard GF-5(02) satellite**

| Spectrum | UV | VIS | | | | NIR | SWIR | | |
|---|---|---|---|---|---|---|---|---|---|
| Central band (nm) | 381 | 410 | 442 | 489 | 670 | 865 | 1380 | 1611 | 2254 |
| Polarization | I/Q/U | | | | | | | | |
| Expected accuracy | ΔI 5% (UV-NIR-NIR); ΔI 6% (SWIR); ΔDoLP 0.005 | | | | | | | | |
| L1B Spatial Sampling | Nadir: 6.4 km; Edge: 25 km | | | | | | | | |
| Full Width Half Maximum  (nm) | 20 | | | | | 40 | | 60 | 80 |
| Rotational period (sec) | 0.97911 | | | | | | | | |
| Scanning width | 2110 km (+/- 53°) | | | | | | | | |
| Orbit | 706 km; Equatorial crossing time 10:30 L. T. (descending node) | | | | | | | | |

Our development of the Level 2 aerosol and surface processing starts from the POSP Level 1B TOA measurements. Before

re-griding Level 1B onto Level 1C 10 km x 10 km grid, we use 3 consecutive steps to filter Level 1B data for aerosol and surface retrieval. Figure 1 shows the schematic diagram to illustrate the POSP data preprocessing from Level 1B to Level 1C for aerosol and surface retrieval.

**Step 1**: Skip left and right 10 pixels at the edges of each swath.

120 The spatial resolution of POSP L1B pixel is approximately 6.4 km at nadir, and spatial sampling size become larger as the scan gets closer to the edge of the track (Fig. 1). The POSP field of view is +/- 53°, orbit height is at ~706 km, therefore the pixel size at the edge is approximately 25 km. According to the recommendation from Level 1 data team, we simply skip 10 pixels at the edges of each swath and proceed with 10 km Level 2 aerosol and surface retrieval.

125 **Step 2**: Cloud mask

Although the delivered POSP L1B data come with cloud identification, but some evident misclassifications are observed and it is with limited performance. We apply three additional conditions to filter cloudy pixels based on simply threshold methods. Specifically, TOA reflectance R (1380 nm) < 0.02 is used to filter possible cirrus cloud. Then, the MODIS spatial variability method (3x3 window standard deviation) is used (Martins et al., 2002) and the threshold is slightly adjusted for

130 POSP. We use 3x3_STD_R (412 nm) <0.03 over land, and 3x3_STD_R (865 nm) <0.03 over ocean. In addition, we use a one-pixel buffer zone for possible cloud shadow. Any pixel next to an identified cloudy pixel is removed for aerosol and surface retrieval. Overall, there is a big potential for improving cloud identification for POSP Level 1B data generation, especially the polarimetric measurements are not used for the moment. The snow/ice mask is not used in this study, and part of snow/ice pixels can be removed in the cloud procedure, and others may affect our Level 2 aerosol and surface retrieval.

 **Step 3**: Ocean glint mask

The ocean sun-glint mask is achieved by applying threshold on the glint angle ($\theta_g$) using solar, view zenith angles ($\theta_s$ and $\theta_v$) and their relative azimuth angle ($\Delta\phi$) (see Eq. 1) (Cox and Munk, 1954). Small glint angle indicates higher probabilities of sun-glint. In this study, each ocean pixel with glint angle smaller than 40 degree is filtered.

140 $\cos\theta_g = \cos\theta_v\cos\theta_s - \sin\theta_v\sin\theta_s\cos\Delta\phi$         (1)

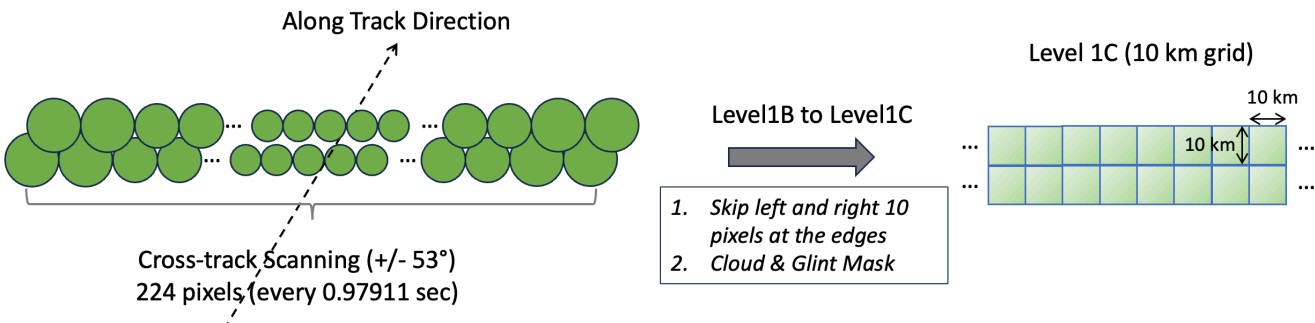

**Figure 1. Schematic diagram to illustrate POSP data preprocessing (Level 1B to Level 1C)**

Figure 2 shows an example of POSP TOA Level 1B measurements on 2022-02-28 (Fig. 2a) and its corresponding Level 1C preprocessed for aerosol and surface retrieval (Fig. 2b). The TOA false color RGB image is composed with POSP reflectance from 670, 489 and 410 nm channels respectively.

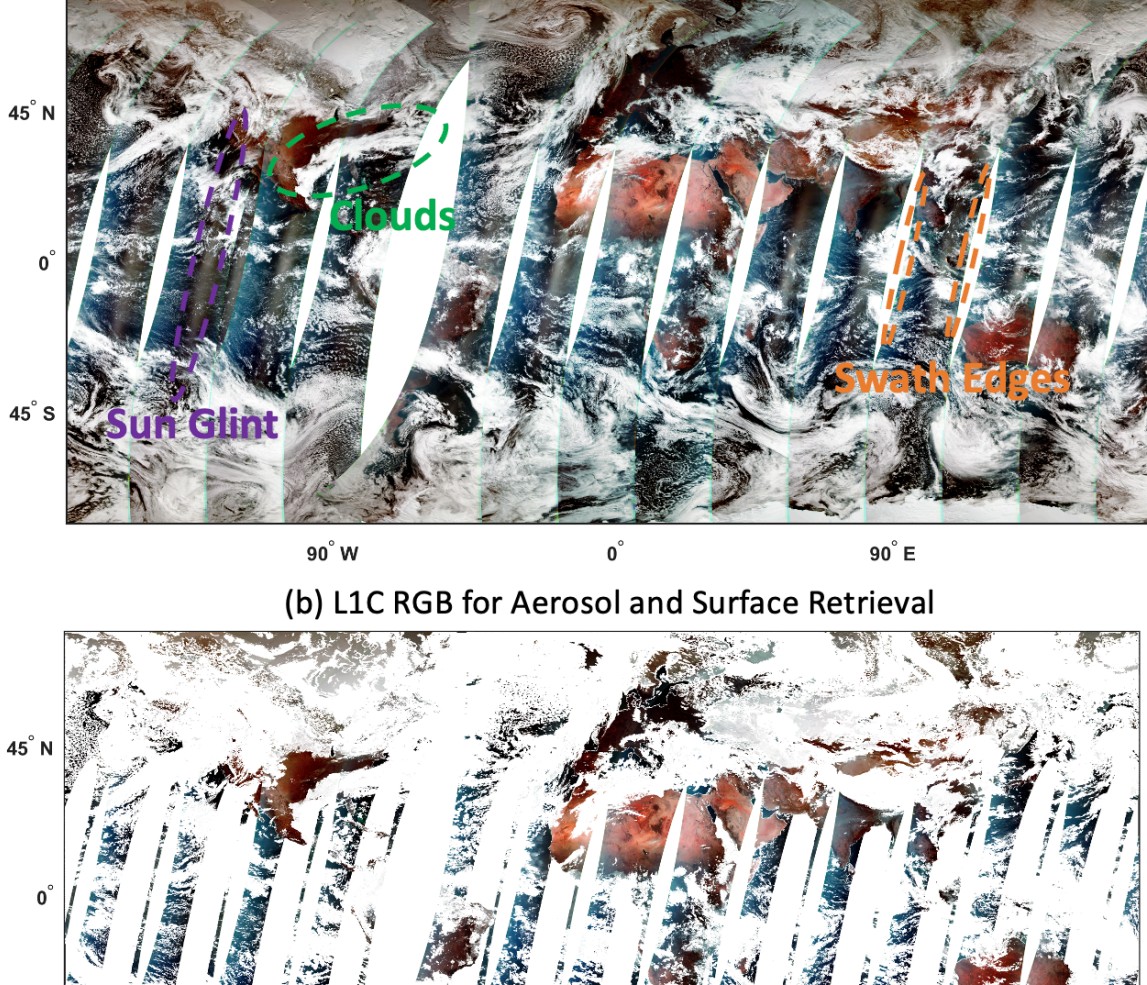

Figure 2. An example POSP TOA measurements on 2022-02-28 to illustrate (a) Level-1B TOA false color RGB and (b) Preprocessed Level-1C data prepared for aerosol and surface retrieval.

## 2.2 Datasets used for validation and evaluation

### 2.2.1 Ground-based AERONET aerosol reference dataset

In order to validate the retrieval of aerosol optical and microphysical properties, the ground-based Aerosol Robotic Network (AERONET) aerosol reference dataset is used (Holben et al., 1998). AERONET direct-Sun multi-spectral aerosol optical depth (AOD), Ångström Exponent, spectral deconvolution algorithm (SDA) fine mode AOD (AODF) and coarse mode AOD (AODC) (O'Neill et al., 2003), as well as single scattering albedo (SSA) from inversion product based on the almucantar measurements (Dubovik et al., 2000; Dubovik and King, 2000; Dubovik et al., 2000, 2002, 2006) are used to

validate POSP aerosol retrievals. Specifically, we use the up-to-date AERONET Version 3 Level 2 products (last access: 2024-02-19) (Giles et al., 2019; Sinyuk et al., 2020; Smirnov et al., 2000) and make use of all AERONET sites with available data during the study period.

### 2.2.2 Space-borne aerosol and surface datasets from VIIRS and MODIS

Space-borne aerosol and surface datasets obtained from NOAA's Visible Infrared Imaging Radiometer Suite (VIIRS) aboard

on the NOAA-20 satellite and NASA's Moderate Resolution Imaging Spectroradiometer (MODIS) aboard on the TERRA and AQUA satellites are used in this study to compare with derived POSP aerosol and surface products. Specifically, the NOAA20 VIIRS Deep Blue (DB) 6 km Version 2.0 Level 2 aerosol product (AERDB_L2_VIIRS_NOAA20) is used to intercompare with the POSP aerosol product (Lee et al., 2024). The AERDB_L2_VIIRS_NOAA20 dataset includes spectral AOD at 412, 490, 550 and 670 nm over land and AOD at 490, 550 and 670 nm over ocean. The fine mode fraction at 550

170 nm is also provided over ocean that can be used to derive AODF and AODC at 550 nm (Hsu et al., 2019; Sayer et al., 2018a, b). For surface properties, the MODIS Collection 6.1 Albedo Model Parameters produced at 0.05° (degree) climate model grid datasets (MCD43C3) are used in the intercomparison including MCD43C3 surface albedos (white sky albedo and black sky albedo) at Band 1 (620-670 nm), Band 2 (841-876 nm), Band 3 (459-479 nm) and Band 7 (2105-2155 nm) (Schaaf and Wang, 2015).

## 3 Aerosol and surface retrieval scheme and product specification

### 3.1 POSP/GF-5(02) Level 2 aerosol and surface retrieval scheme

The GRASP/Models approach is used to retrieve aerosol and surface properties from POSP/GF-5(02) measurements. On the top of GRASP platform, several aerosol modeling approaches were developed by assuming different parameterization scheme for size distribution, complex refractive index, as well as chemical composition (Dubovik et al., 2011, 2021a; Li et

al., 2019; Chen et al., 2020). Among them, "Models" approach is one of the most simplified approaches with 6 spectral independent parameters to represent aerosol optical and microphysical properties. The GRASP/Models approach has been proven to be efficient to retrieve high-quality aerosol properties from different types of satellite remote sensing

measurements, including multi-angular polarimeter, multi-spectral/hyperspectral spectrometer, etc. (Chen et al., 2020, 2022b, 2024a, b; Jin et al., 2022; Litvinov et al., 2024).

Specifically, aerosol total single scattering characteristics are assumed to be linear combination of the scattering characteristics of several pre-defined aerosol models (Eq. 2).

$$\omega(\lambda)\mathbf{P}(\lambda, \Theta) = \frac{\sum_{k=1}^{n} c_k C_{ext}^k(\lambda)\omega_k \mathbf{P}_k(\lambda, \Theta)}{\sum_{k=1}^{n} c_k C_{ext}^k(\lambda)} \tag{2}$$

Where $\lambda$ indicates spectral channel; $\Theta$ is scattering angle; $\omega$ is aerosol single scattering albedo (SSA); $\mathbf{P}$ denotes phase

matrix; $k$ denotes one of $n$ aerosol models; $C_{ext}$ denotes total aerosol extinction; $c_k$ denotes relative fraction for $k$-th aerosol model. Following previous applications on OLCI/Sentinel-3 and TROPOMI/Sentinel-5p (Chen et al., 2022b, 2024a; Litvinov et al., 2024), we use 4 models ($n$=4) approach to process POSP/GF-5(02), $k$=1 fine absorbing (biomass burning), $k$=2 fine slightly absorbing (urban/sulphate aerosol), $k$=3 coarse maritime (sea salt), $k$=4 coarse dust. The definition of 4 models' size distribution, spectral complex refractive index and non-sphericity are based on the AERONET climatology

(Dubovik et al., 2002; Lopatin et al., 2021).

In addition, taking into account of the limited information from passive remote sensing measurements, aerosol microphysical properties are assumed to be identical at different vertical layers and only the total aerosol volume concentration and aerosol total extinction profile ($c_v = \tau_{ext}(\lambda)/\sum_{k=1}^{4} c_k C_{ext}^k(\lambda)$) varies following the exponential function.

$$c_v(z) = \frac{c_v}{h}\exp\left(-\frac{z}{h}\right) \tag{3}$$

Where $z$ represents atmosphere vertical level; $h$ is the aerosol layer height, namely scale height.

The sum of the semi-empirical Ross-Li sparse Bidirectional Reflectance Function (BRF) model (Li and Strahler, 1992; Roujean et al., 1992; Wanner et al., 1995) and the polarized reflection matrix $R_p$ based on semi-empirical Maignan-Bréon

BPDF (Bidirectional Polarization Distribution Function) models (Litvinov et al., 2010, 2011a, b; Maignan et al., 2009) are used to model surface reflectance for POSP/GF-5(02) processing. The kernel-driven Ross-Li BRDF model uses a linear combination of three kernels $f_{iso}$, $f_{vol}$ and $f_{geom}$ representing isotropic, volumetric and geometric optics surface scattering respectively (Li and Strahler, 1992; Roujean et al., 1992; Wanner et al., 1995). In addition, according to Litvinov et al. (Litvinov et al., 2010, 2011a, b), this model is further renormalized to spectral invariant of the second ($a_{vol}$) and third

($a_{geom}$) BRDF parameters in comparison to the first one $a_{iso}(\lambda)$. In GRASP, the polarized reflection matrix $R_p$ is based on the semi-empirical Maignan-Bréon BPDF model (Maignan et al., 2009). For POSP/GF-5(02) processing, the wavelength independent scaling parameter ($\alpha$) for Fresnel-based reflection matrix is retrieved.

The ocean surface reflectance is modelled as combination of different ocean surface approaches. The Fresnel's reflection from the sea surface is taken into account using the Cox and Munk model (Cox and Munk, 1954). The water leaving radiance is assumed to be isotropic and is taken into account by Lambertian unpolarized reflectance (Voss et al., 2007). The relations between whitecaps/foam fraction and wind speed are defined by the empirical formula by Monahan and O'Muircheartaigh (1980). The spectral dependence of foam is accounted with the model described in (Frouin et al., 1996; Frouin and Pelletier, 2015). The detailed description of ocean surface reflectance modeling approach used in GRASP approach can be found in Litvinov et al. (2024) and Chen et al. (2022b). The spectral dependent isotropic water-leaving radiance $r_0(\lambda)$, Fresnel reflection fraction (free from foam, dense sediments, whitecaps etc.) $\delta_{Fr}$ and the mean square facet slope $\sigma^2$ are the directly retrieved parameters for ocean surface.

Then the state vector of each land and ocean pixel for POSP/GF-5(02) processing with GRASP/Models approach can be represented as follows:

$$\mathbf{X}^{land} = [c_v, c_1, c_2, c_3, c_4, h, a_{iso}(\lambda), a_{vol}, a_{geom}, \alpha]^T \tag{4}$$

$$\mathbf{X}^{ocean} = [c_v, c_1, c_2, c_3, c_4, h, r_0(\lambda), \delta_{Fr}, \sigma^2]^T \tag{5}$$

Where $c_v$ is columnar aerosol volume concentration; $c_{1...4}$ is volume fractions of four different aerosol models; $h$ is aerosol scale layer height; $a_{iso}(\lambda)$ is linear coefficient of the Ross-Li BRDF spectral dependent isotropic kernel; $a_{vol}$ and $a_{geom}$ are the linear coefficients of the spectral invariant volumetric and geometric terms; $\alpha$ is the Maignan-Bréon BPDF model spectral independent scaling parameter for Fresnel-based reflection matrix. $r_0(\lambda)$ is spectral dependent isotropic water leaving reflectance; $\delta_{Fr}$ is the fraction of surface which provides Fresnel reflection (free from foam, dense sediments, whitecaps etc.); $\sigma^2$ the mean square facet slope.

The inhomogeneous POSP 10 km pixels partially coved by water are skipped for the processing. Based on the directly retrieved parameters in state vector, aerosol optical-microphysical properties and surface properties, such as AOD, AODF, AODC, AAOD, SSA, NDVI, BRDF, BPDF, different surface albedos (White-Sky Albedo, Black-Sky Albedo) at UV, VIS, NIR and SWIR spectrum, can be calculated and delivered as a product.

The GRASP numerical inversion is implemented as a statistically optimized fitting of the observations using the least-square multi-Term principle (Dubovik et al., 2021a). GRASP implements a multi-pixel technique to improve retrieval by simultaneously inverting large group of pixels (Dubovik et al., 2011, 2014). This approach allows for significant enhancement of atmosphere properties retrievals from remote sensing measurements by means of using additional a priori information about "correlation" retrieved properties in different pixels of the inverted group. The smallest processing group (segment) in the POSP/GRASP scheme is a group of 3x3xNT pixels, where NT is the number of measurements available in a month.

### 3.2 POSP/GF-5(02) Level 2 aerosol and surface products specification

The primary POSP/GF-5(02) aerosol product is the spectral AOD, which measures the column amount of aerosols in the atmosphere and is widely used in the community. POSP/GRASP Level 2 product offers spectral AOD at 9 wavelengths ranging from UV to SWIR. The product includes several additional aerosol characteristics such as AE, spectral fine/coarse mode AOD, spectral SSA and aerosol scale height. These specific aerosol characteristics are connected to aerosol particle size and absorption properties, which are important criteria for the aerosol climate and air quality community. The POSP/GRASP also offers full surface BRDF and BPDF characteristics, as well as the surface NDVI, spectral Directional Hemispherical Reflectance (DHR or black sky albedo) and spectral Surface Isotropic Bihemispherical Reflectance (BHR$_{ISO}$ or white sky albedo). The POSP/GF-5(02) Level 2 aerosol and surface products are contained in a file typically named as: *GF5B_POSP_L2_Land_Ocean_{YYYYMMDD}.nc*. The product fields contained in the NetCDF-4 product file are listed in the Table 2.

**Table 2. List of fields in the POSP/GF-5(02) Level 2 aerosol and surface product.**

| Variable Name | Dimension | LongName and Description |
|---|---|---|
| aaod{$\lambda$} | (y, x, $\lambda$) | Aerosol Absorption Optical Depth |
| AeroModelCon1 | (y, x) | Aerosol fine absorbing mode fraction |
| AeroModelCon2 | (y, x) | Aerosol fine non-absorbing mode fraction |
| AeroModelCon3 | (y, x) | Aerosol oceanic mode fraction |
| AeroModelCon4 | (y, x) | Aerosol dust mode fraction |
| AerosolConcentration | (y, x) | Aerosol total volume concentration |
| AExp | (y, x) | Ångström Exponent (442/865) |
| bhr_iso{$\lambda$} | (y, x, $\lambda$) | Surface Isotropic Bihemispherical Reflectance |
| dhr{$\lambda$} | (y, x, $\lambda$) | Directional Hemispherical Reflectance |
| land_percent | (y, x) | Percentage of land |
| LandBPDFMaignanBreon | (y, x) | Fresnel-based reflection matrix scaling parameter |
| LandBRDFRossLi{$\lambda$}_1 | (y, x, $\lambda$) | Ross Li BRDF isotropic parameter |
| LandBRDFRossLi_2 | (y, x) | Ross Li BRDF normalized volumetric parameter |
| LandBRDFRossLi_3 | (y, x) | Ross Li BRDF normalized geometric parameter |
| lat | (y, x) | latitude |
| Lon | (y, x) | longitude |
| NDVI | (y, x) | Normalized Difference Vegetation Index |
| residual_absolute0 | (y, x) | Fitting Residual (absolute) of measured reflectance |
| residual_relative0 | (y, x) | Fitting Residual (relative) of measured reflectance |

| residual_absolute1 | (y, x) | Fitting Residual (absolute) of measured DOLP |
|---|---|---|
| residual_relative1 | (y, x) | Fitting Residual (relative) of measured DOLP |
| ssa{$\lambda$} | (y, x, $\lambda$) | Single Scattering Albedo |
| tau{$\lambda$} | (y, x, $\lambda$) | Aerosol Optical Depth |
| tauC{$\lambda$} | (y, x, $\lambda$) | Coarse mode Aerosol Optical Depth |
| tauF{$\lambda$} | (y, x, $\lambda$) | Fine mode Aerosol Optical Depth |
| unixtimestamp | (y, x) | Unix time in seconds from 1970/01/01T00:00:00 (UTC +8 hours) |
| VertProfileHeight | (y, x) | Scale height of vertical profile |
| WaterBRMCoxMunkIso{$\lambda$}_1 | (y, x, $\lambda$) | Surface albedo of water body |
| WaterBRMCoxMunkIso_2 | (y, x) | Cox Munk BRDF fresnel fraction |
| WaterBRMCoxMunkIso_3 | (y, x) | Cox Munk BRDF msq slope |
| X | x | WGS84 coordinates (easting) |
| Y | y | WGS84 coordinates (northing) |

$\lambda$ = 381, 410, 442, 489, 670, 865, 1611, 2254 nm

## 4 Preliminary validation and evaluation of POSP/GF-5B Level 2 product

### 4.1 Global distribution of POSP/GF-5(02) Level 2 aerosol and surface product

Based on the method proposed in Section 3, we processed first 18 months of POSP/GF-5(02) measurements from December 2021 to May 2023 using 4 intel 32-cores Think-Station. Figure 3 shows the spatial distribution of valid retrievals from December 2021 to May 2023. In summary, approximately 113 million pixels are retrieved successfully, account 70 million land pixels (~62%) and 43 million ocean pixels (~38%). Evidently, the ocean pixels percentage is much lower than expected, which can be associated with too strict cloud mask and glint mask used over ocean. This need to be adjusted and tested for future processing. We estimated the retrieval speed in order to constrain computational cost for future operational processing based on the 18 months processing exercise. Approximately, for land pixels, the mean speed is 0.32 second / pixel / core with $1\sigma = 0.12$ sec/pixel/core; for ocean pixels, the mean speed is 0.38 second / pixel / core with $1\sigma = 0.13$ sec/pixel/core.

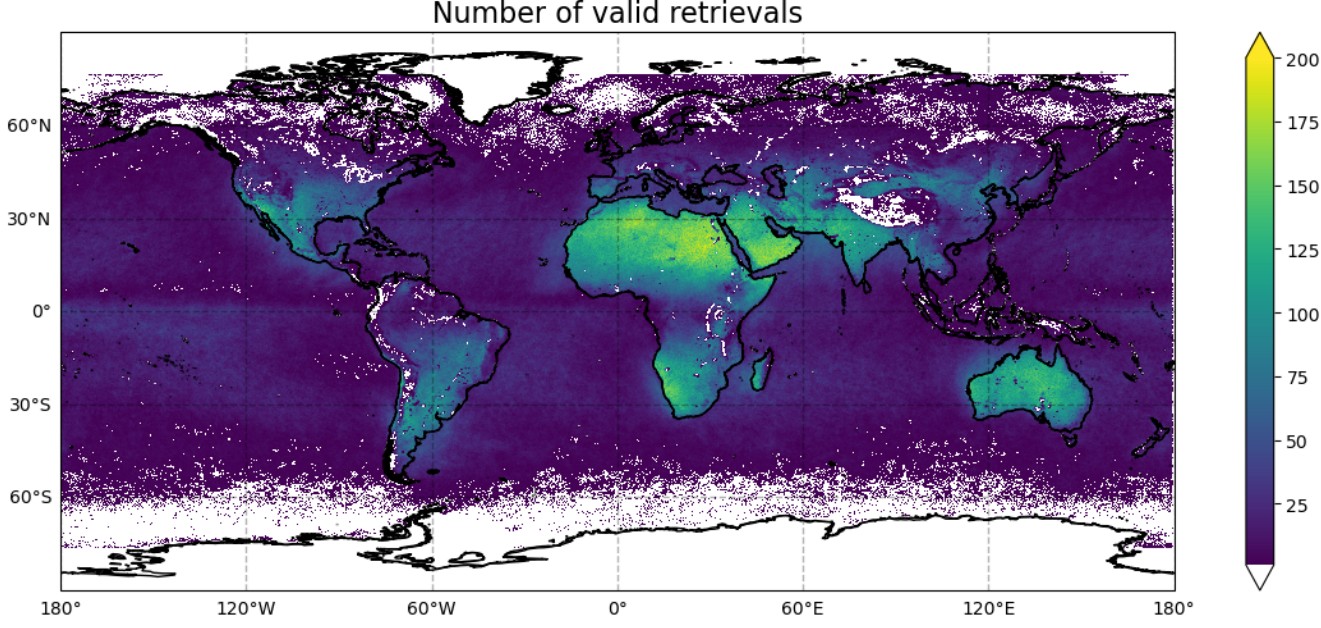

**Figure 3. Spatial distribution of number of valid POSP/GRASP retrievals from December 2021 to May 2023.**

In order to have a general picture of the POSP/GF-5(02) aerosol and surface product, we present the spatial distribution of POSP/GF-5(02) aerosol and surface main characteristics in Fig. 4 (main aerosol products) and Fig. 5 (main surface products)

from December 2021 to May 2023. Figures 4 and 5 show the mean of main aerosol and surface products of the entire 18 month of data. As for the detailed aerosol properties, such as SSA (550 nm), AE (440/870) and Scale Height, we select the valid retrieval with POSP AOD (550 nm)>0.2 on daily basis and then calculate the mean values shown in Figs. 4d-f because these the retrieval accuracy of these detailed parameters is strongly depending on the aerosol information content, simplistically aerosol loading/AOD. The main POSP/GF-5(02) aerosol products include spectral AOD, AODF, AODC and

SSA range covering UV, VIS, NIR and SWIR spectrum, as well as AE (440/870), aerosol scale height (ALH). Note spectral aerosol absorption optical depth (AAOD) can be derived using $AAOD(\lambda) = AOD(\lambda) \times (1 - SSA(\lambda))$ and is also provided in the product file. The main POSP/GF-5(02) surface products include full Ross Li BRDF parameters $(a_{iso}(\lambda), a_{vol}, a_{geom})$, Maignan-Bréon BPDF Fresnel-based reflection matrix scaling parameter $(\alpha)$, surface white sky albedo BHRiso$(\lambda)$, black sky albedo DHR$(\lambda)$, as well as surface Normalized Difference Vegetation Index (NDVI).

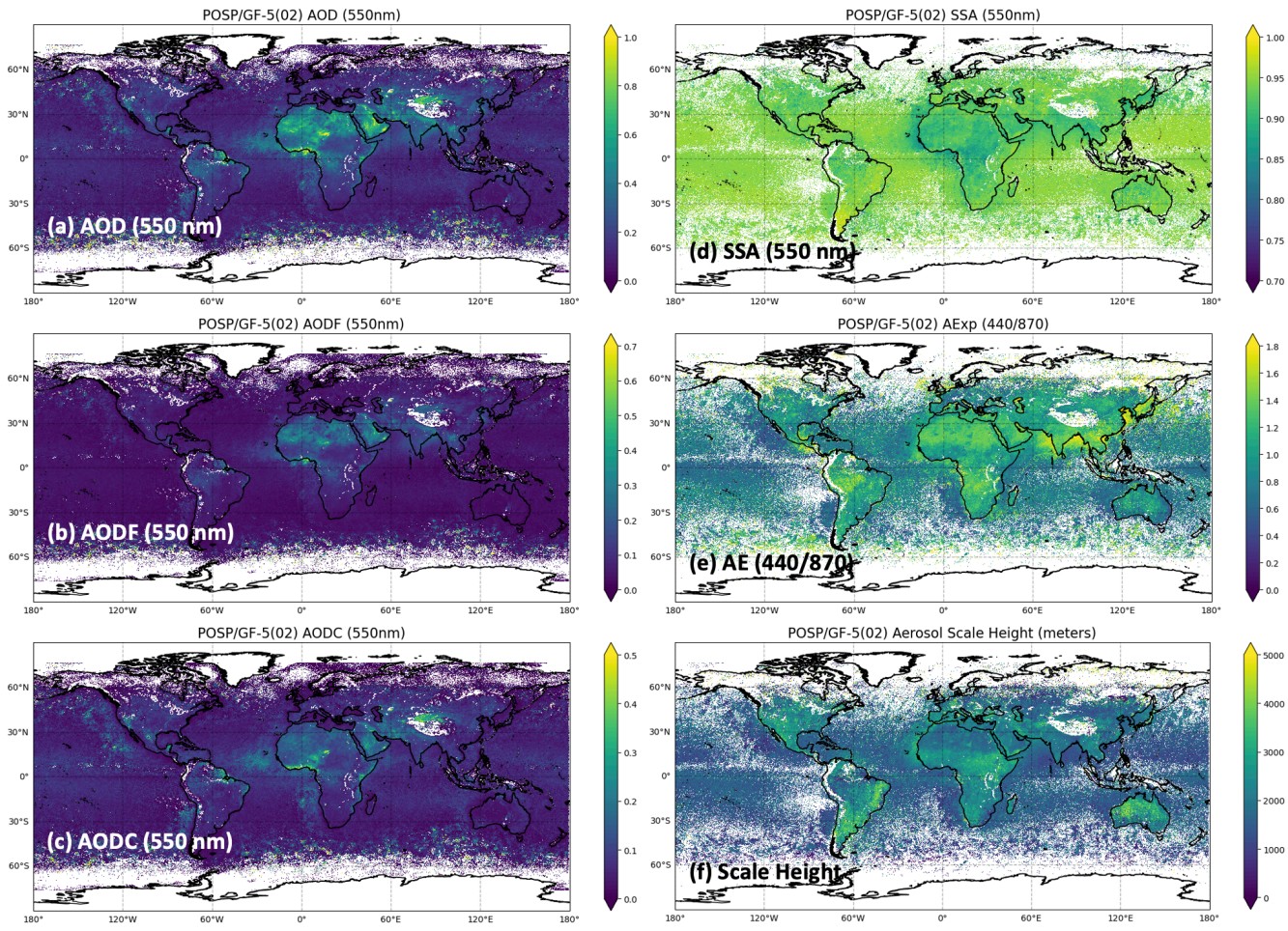

**Figure 4. Spatial distribution of POSP/GF-5(02) main aerosol products: (a) Aerosol Optical Depth – AOD; (b) Fine mode Aerosol Optical Depth – AODF; (c) Coarse mode Aerosol Optical Depth (AODC); (d) Single Scattering Albedo – SSA; (e) Ångström Exponent – AE (440/870); (f) Scale height of aerosol vertical profile – ALH; Note AOD, AODF, AODC, SSA are spectral dependent and provided at UV, VIS, NIR and SWIR spectrum. The values represent the**

**mean of each characteristic over the entire 18-month of POSP data (December 2021 to May 2023). The AOD (550 nm) > 0.2 criteria is used to select high-quality SSA (550 nm), AE (440/870) and ALH on daily basis before calculating its mean values.**

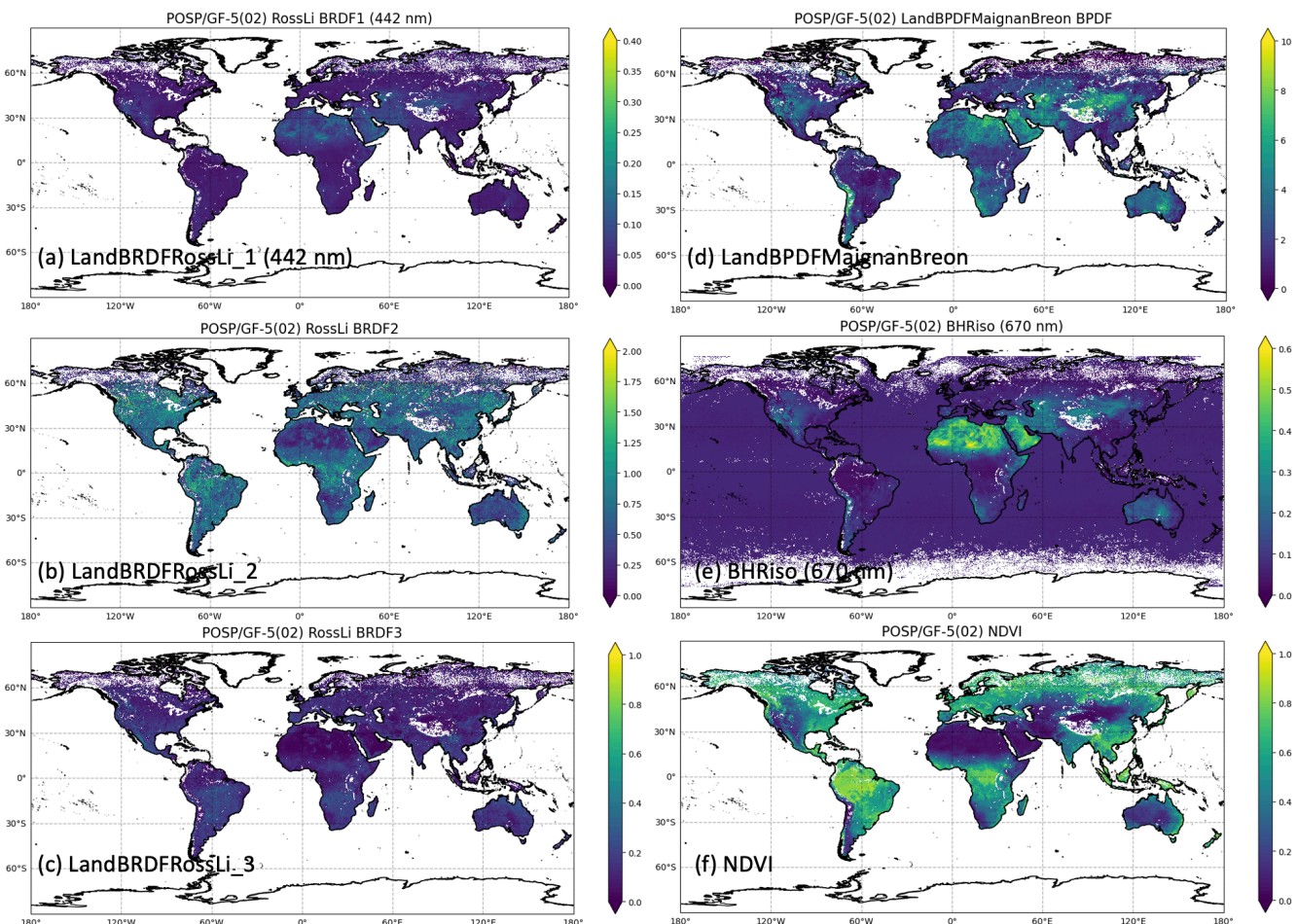

Figure 5. Spatial distribution of POSP/GF-5(02) main surface products: (a) Ross Li BRDF isotropic parameter; (b) Ross Li BRDF normalized volumetric parameter; (c) Ross Li BRDF normalized geometric parameter; (d) Maignan-Bréon BPDF; (e) Surface Isotropic Bihemispherical Reflectance – BHRiso or White Sky Albedo; (f) Normalized Difference Vegetation Index – NDVI. Note BRDF1 and BHRiso (White Sky Albedo) are spectral dependent and provided at UV, VIS, NIR and SWIR spectrum. The values represent the mean of each characteristic over the entire 18-month of POSP data (December 2021 to May 2023).

During the first POSP/GF-5(02) processing, we also identify some remain issues in the current baseline Level 2 products. (i) The cloud and glint mask over ocean seems too strict resulting in the ocean pixels percentage is much lower than expected; (ii) We also identify some existing stripes in the Level 2 aerosol and surface products that are caused by the delay of Level 1 data production, therefore some Level 1 tracks are recorded until the next day and overwrite the coming data. These known issues are documented in the data description and expected to be solved in the next processing.

## 4.2 Validation of POSP/GF-5(02) aerosol product with AERONET

In order to verify the obtained spatial distribution of POSP aerosol and surface products, POSP aerosol product is validated with ground-based AERONET dataset. In order to match satellite retrievals with AERONET measurements, we follow the strategy used in our previous studies (Chen et al., 2020, 2022b, 2024a). Specifically, we use a 3 x 3 window centered over AERONET sites, and the satellite retrievals with a 3 x 3 window is averaged. Meanwhile, the available AERONET direct-Sun AOD, AE, AODF and AODC products are averaged within +/- 30 min of the POSP/GF-5(02) satellite overpass, and AERONET inversion SSA products are averaged within +/- 180 min of the POSP/GF-5(02) satellite overpass. For POSP, we have not yet introduced the quality flags into the product as for TROPOMI/Sentinel-5p (Litvinov et al., 2024). While, some experiences from TROPOMI/Sentinel-5p quality flag definition are used to select high-quality POSP retrievals for validation with AERONET. Basically, any retrieval meets the 3 conditions is defined as "high-quality" and will be used in validation.

(i) The relative fitting residual (residual_relative_noise0) smaller than 5%;

(ii) Over land, the AOD (865 nm) standard deviation within its 3x3 window (STD_AOD$_{865}$) smaller than 0.1 or STD_AOD$_{865}$ / $\overline{AOD_{865}}$ smaller than 0.2, where $\overline{AOD_{865}}$ is the mean AOD with the 3x3 window. Over ocean, the STD_AOD$_{865}$ is smaller than 0.05;

(iii) The number of valid retrievals with a 3x3 window need to be higher or equal to 3.

To quantify the validation results, several standard statistical parameters are used in this study, including linear correlation coefficient (R), the root mean square error (RMSE), the bias (BIAS), the fulfilment of the AOD Global Climate Observation System requirement (GCOS fraction), the formulated Target requirement for AOD, and the formulated Optimal and Target requirements for AE, SSA and surface albedos. Note the Optimal and Target requirements are formulated in our previous ESA S5P+I AOD/BRDF project (Litvinov et al., 2024) and were used in TROPOMI/Sentinel-5p validation (Chen et al., 2024a). Specifically, the GCOS requirement for AOD, AODF and AODC is max 0.04 or 10% (whatever is bigger); the Optimal requirements for AE, SSA and surface albedos are 0.3, 0.03 and 0.01 respectively; the Target requirement for AOD, AODF and AODC is max 0.05 or 20% (whatever is bigger), and the Target requirement for AE, SSA and surface albedos are 0.5, 0.05 and 0.02 respectively.

Figure 6 shows the validation of an entire year 2022 POSP/GF-5(02) GRASP AOD (550 nm) with AERONET over land (Fig. 6a – linear scale; Fig. 6b – logarithmic scale) and ocean (Fig. 6c – linear scale; Fig. 6d – logarithmic scale). Generally, POSP AOD (550 nm) show good agreement with AERONET both over land and ocean, with R=0.813, RMSE=0.114 over land and R=0.861, RMSE=0.091 over ocean. Moreover, the fulfilment of GCOS requirement is 47.6% over land and 54.3% over ocean. We intercompare the validation results obtained from TROPOMI/GRASP one year validation (Chen et al., 2024a; Litvinov et al., 2024). The obtained POSP validation statistical metrics are close to the TROPOMI validation results for QA>=2, where the fulfilment of AOD GCOS requirement is 48.4% over land and 71.3% over ocean (Litvinov et al.,

2024). Besides, the matchup points are evidently less than TROPOMI, which is associated with POSP's small swath width (1850 km) relative to TROPOMI (2600 km), and the scanning polarimeter has a large deformation at the edge of the track that we have to remove 10 pixels at the edge of the track.

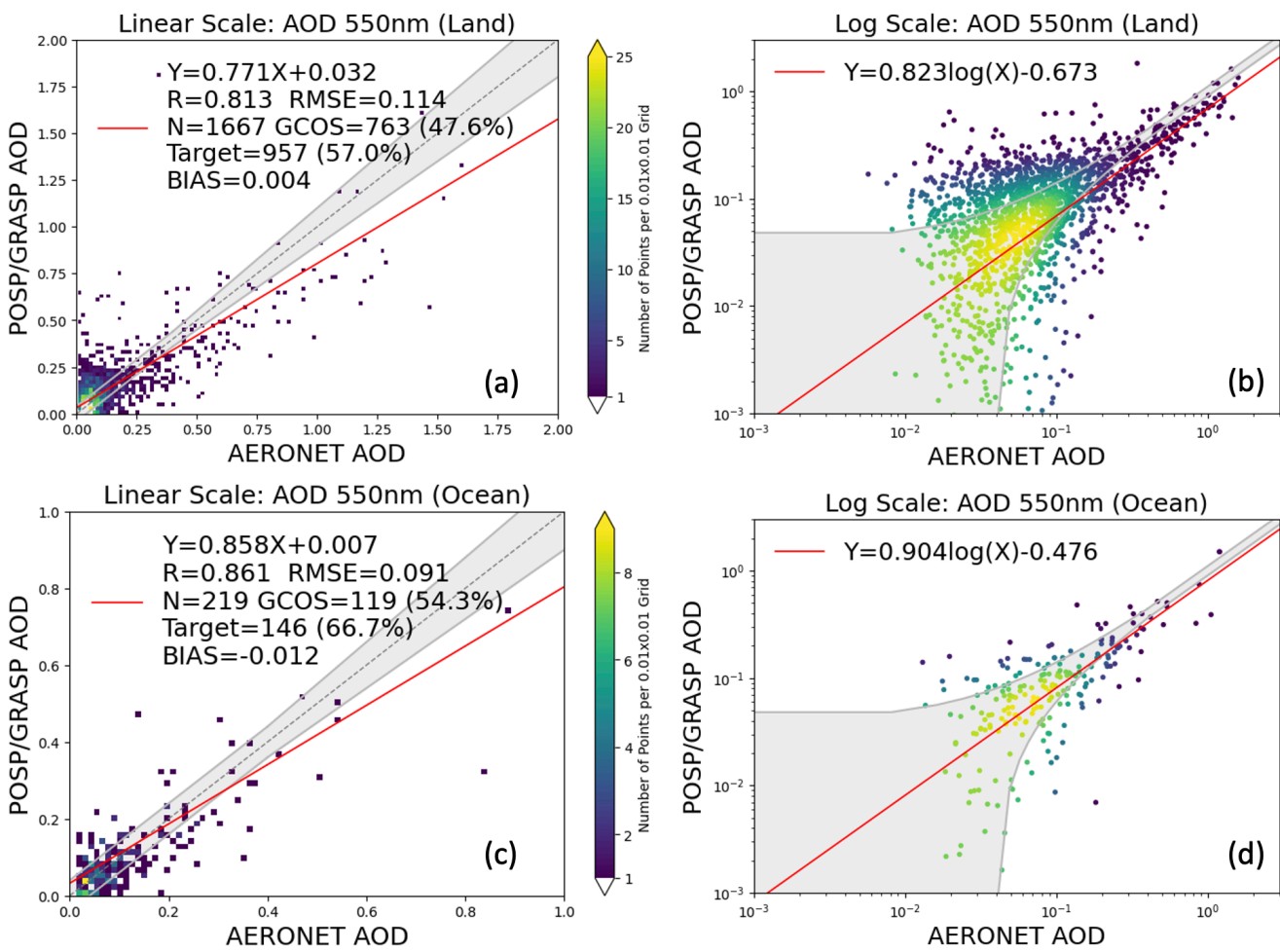

**Figure 6. Validation of POSP/GF-5(02) GRASP AOD (550 nm) with AERONET over land and ocean for an entire year 2022. (a) POSP AOD validation with linear scale over land; (b) POSP AOD validation with logarithmic scale over land; (c) POSP AOD validation with linear scale over ocean; (d) POSP AOD validation with logarithmic scale over ocean.**

The validation of POSP/GF-5(02) GRASP AE (440/870 nm) with AERONET is present in Figure 7 (7a – Land; 7b – Ocean). In order to ensure the quality of the satellite AE (440/870 nm) product, we usually filter low AOD cases for AE

validation, due to the fact that the calculated AE is very sensitivity to the small variations in AOD at different wavelengths. For POSP AE validation, we select cases with POSP AOD (550 nm) > 0.2 over land and POSP AOD (550 nm) > 0.2 over

355 ocean. Generally, POSP/GRASP tends to underestimate AE (too large particle size) for small particles (AE>1.0) and overestimate AE (too small particle size) for large particles (AE<1.0) over land. This is similar to the results obtained from OLCI and POLDER using GRASP/Models approach, and is associated with the particle size assumption for aerosol models. Moreover, the obtained AE>1.0 over Sahara Desert (Fig. 4e) seems to be problematic, and is possibly related with polarimetric calibration or SWIR channel radiometric calibration that need further investigations. Over ocean, POSP/GRASP

tends overestimate AE (too small particle size) for all ranges of AE. The results are consistent with the AODF and AODC validation presented in Figure 8 (AODF 550 nm) and Figure 9 (AODC 550 nm). Specifically, we observe a clear tendency over ocean that POSP/GRASP slightly overestimate fine mode AOD (linear scale – Fig. 8c; logarithmic scale – Fig. 8d) while significantly underestimate coarse mode AOD (linear scale – Fig. 9c and logarithmic scale – Fig. 9d). The validation of TROPOMI products show quiet similar feature (Chen et al., 2024a), which can be related with the fact that the space-

borne downward measurements are more sensitive to suspended fine particles than coarse particles near the surface, while the ground-based upward measurements have much more sensitivity to coarse particles near the ground. This is our hypothesis that require more studies in the future.

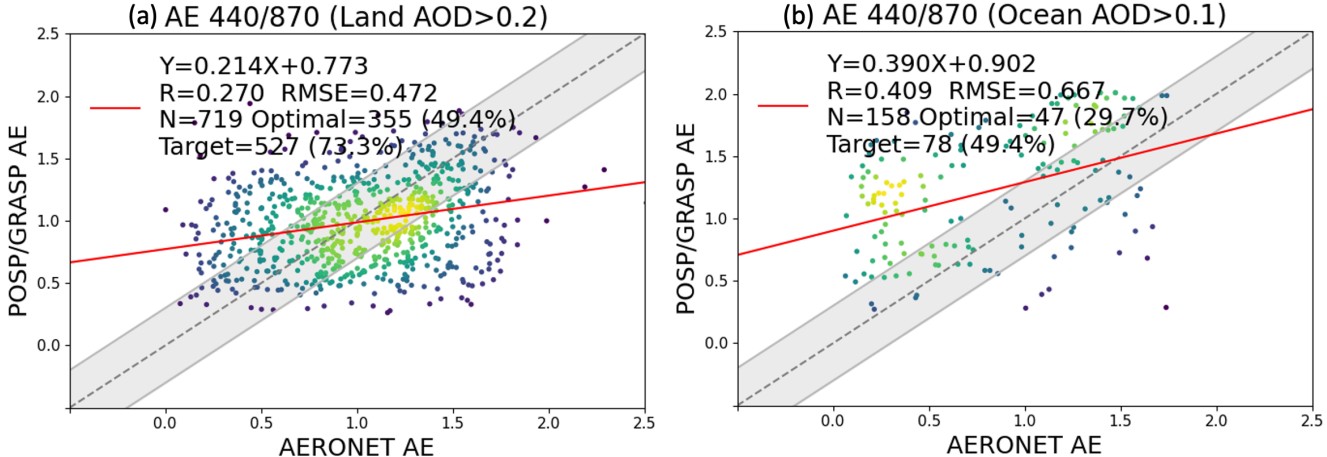

**Figure 7. Validation of POSP/GF-5(02) GRASP AE (440/870) with AERONET over (a) land and (b) ocean for an entire year 2022.**

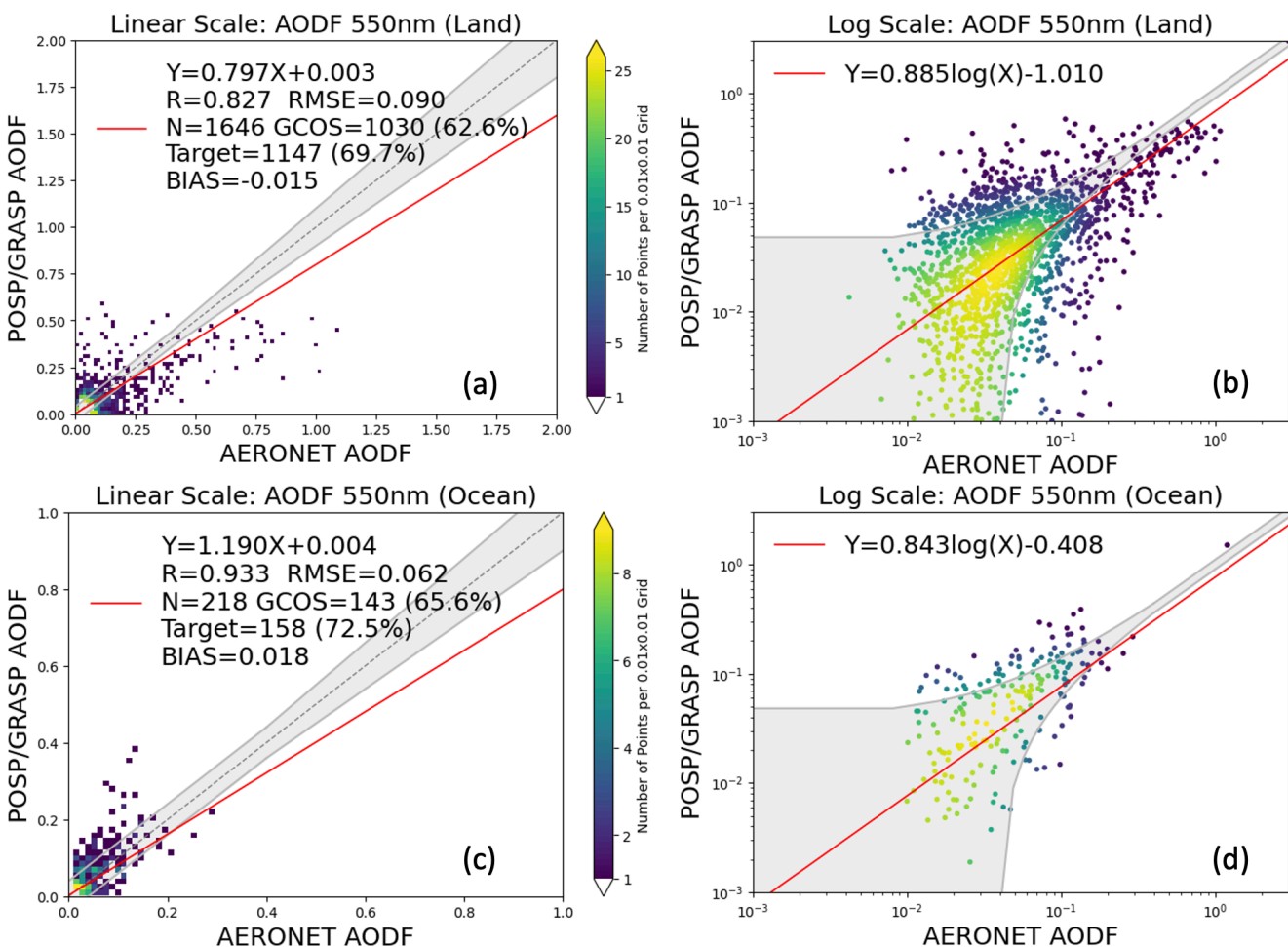

**Figure 8. Validation of POSP/GF-5(02) GRASP AODF (550 nm) with AERONET over land and ocean for an entire year 2022. (a) POSP AODF validation with linear scale over land; (b) POSP AODF validation with logarithmic scale over land; (c) POSP AODF validation with linear scale over ocean; (d) POSP AODF validation with logarithmic scale over ocean.**

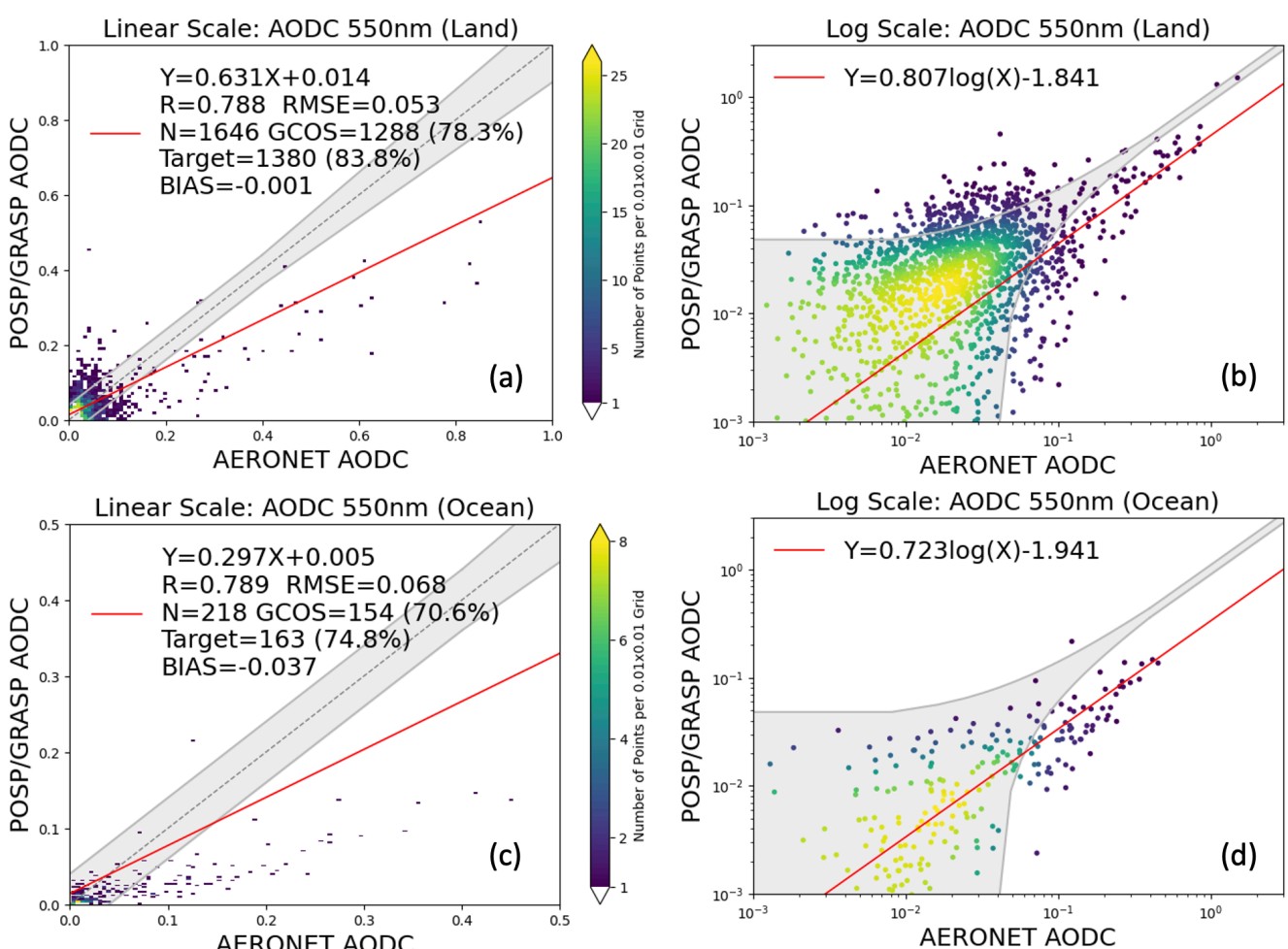

**Figure 9. Validation of POSP/GF-5(02) GRASP AODC (550 nm) with AERONET over land and ocean for an entire year 2022. (a) POSP AODC validation with linear scale over land; (b) POSP AODC validation with logarithmic scale over land; (c) POSP AODC validation with linear scale over ocean; (d) POSP AODC validation with logarithmic scale over ocean.**

In addition, we validate POSP/GF-5(02) GRASP SSA (550 nm) with AERONET inversion products for an entire year 2022 and the results are shown in Fig. 10a, and the validation of POSP/GF-5(02) GRASP SSA spectral dependence, dSSA(440–670) = SSA(440 nm) – SSA(670 nm), are shown in Fig. 10b. The AERONET Level 2 inversion SSA product is used as a reference where it contains aerosol loading filtering (AERONET AOD 440nm>0.4). For POSP/GRASP SSA, we also select the cases with POSP AOD (550 nm) > 0.2 to ensure the retrieval accuracy. POSP/GRASP SSA show good consistency with

AERONET that RMSE is 0.040, BIAS is -0.007 and the fulfilment of SSA Optimal (+/- 0.03) and Target (+/- 0.05) requirements are 63.0% and 81.2% respectively. Spectral dependence of SSA provides important information about the absorption properties and size related parameters. AERONET ground-based dSSA is used to infer and quantify fine/coarse mode absorption components (BC – Black Carbon, BrC – Brown Carbon and DD – Desert Dust) and non-absorbing component, such as ammonia-sulfate (Li et al., 2013; Wang et al., 2013). Due to the general difficulty in retrieving the proper SSA spectral dependence, dSSA is rarely provided by satellite products, therefore seldom appears in satellite product validation. Previously, TROPOMI/Sentinel-5p GRASP product demonstrates good performance to capture the SSA spectral dependence, dSSA(440-670), with 75.8% of correct cases (Litvinov et al., 2024). Here, POSP/GF-5(02) GRASP dSSA also shows reasonable correspondence with ground-based AERONET product with 65.5% of successful separation cases.

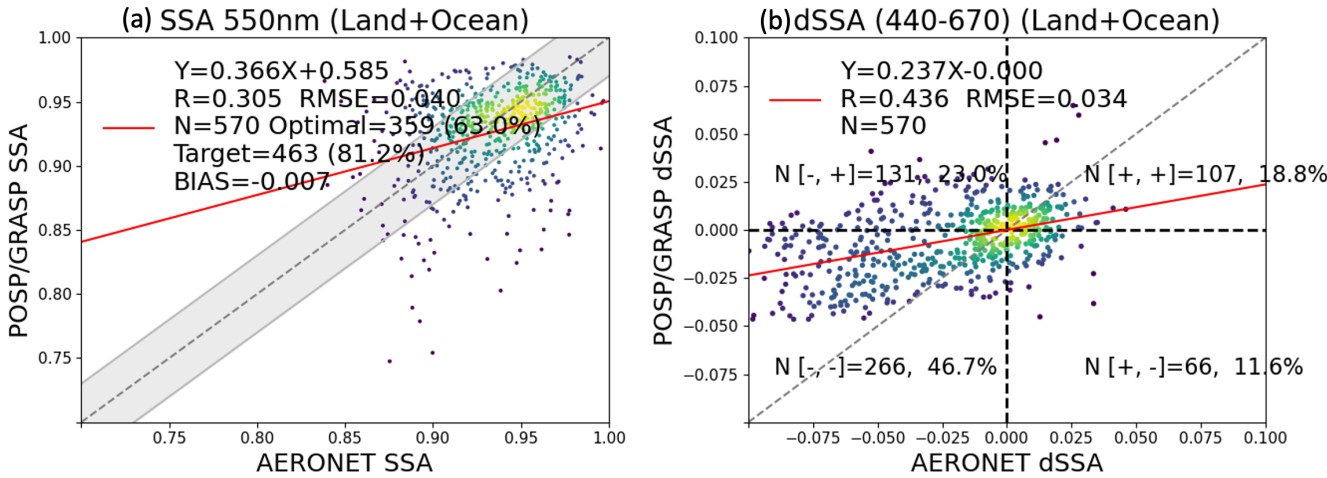

**Figure 10. Validation of POSP/GF-5(02) GRASP (a) SSA (550 nm) and (b) dSSA (440-670) with AERONET over land and ocean for an entire year 2022.**

**Table 3. Intercomparisons of OLCI/Sentinel-3A, TROPOMI/Sentinel-5p and POSP/GF-5(02) AOD (550 nm), AE (OLCI and POSP: 440/870; TROPOMI: 412/670) and SSA (550 nm) one-year validation metrics with AERONET reference dataset. The best performance metric is indicated in bold.**

|  | Sensors (num. pairs) | R | RMSE | BIAS | Optimal% | Target% |
|---|---|---|---|---|---|---|
| AOD 550 nm (Land) | OLCI (3205) | 0.870 | 0.090 | -0.01 | 47.9 | - |
|  | TROPOMI (**7732**) | **0.885** | **0.078** | 0.01 | **56.9** | **67.9** |
|  | POSP (1667) | 0.813 | 0.114 | **0.004** | 47.6 | 57.0 |
| AOD 550 nm (Ocean) | OLCI (217) | 0.872 | **0.047** | **0.01** | 67.3 | - |
|  | TROPOMI (**880**) | **0.881** | 0.047 | **0.01** | **71.3** | **81.1** |

| | | | | | | |
|---|---|---|---|---|---|---|
| | POSP (219) | 0.861 | 0.091 | -0.012 | 54.3 | 66.7 |
| AE (Land) | OLCI (611) | 0.526 | 0.531 | -0.35 | - | - |
| | TROPOMI (**991**) | **0.725** | **0.418** | **0.18** | **51.6** | **76.0** |
| | POSP (719) | 0.270 | 0.472 | - | 49.4 | 73.3 |
| AE (Ocean) | OLCI (46) | **0.751** | 0.386 | 0.18 | - | - |
| | TROPOMI (**1417**) | 0.474 | 0.474 | **0.16** | **52.1** | **71.3** |
| | POSP (158) | 0.409 | 0.667 | - | 29.7 | 49.4 |
| SSA 550 nm (Land + Ocean) | OLCI (115) | 0.471 | **0.026** | -0.01 | - | - |
| | TROPOMI (358) | **0.522** | **0.026** | **0.00** | **82.1** | **94.4** |
| | POSP (**570**) | 0.305 | 0.040 | -0.007 | 63.0 | 81.2 |

Table 3 summarizes the OLCI/Sentinel-3A, TROPOMI/Sentinel-5p and POSP/GF-5(02) products generated by the GRASP algorithm one-year AOD, AE and SSA one-year validation metrics with AERONET reference dataset. The TROPOMI AE (412/670) is used, while OLCI and POSP AE is calculated from 440 to 870 nm. AOD and SSA at 550 nm validation results are used for intercomparison. Note, the one-year validation is performed for different years, for example, OLCI (June 2018 to May 2019), TROPOMI (March 2019 to February 2020) and POSP (January 2022 to December 2022). The OLCI/Sentinel-3A validation metrics are adopted from Chen et al. (2022), TROPOMI/Sentine-5p validation results are obtained from Chen et al. (2024a) and Litvinov et al. (2024) in which QA=3 is used for AOD (550 nm) over land, QA>=2 for AOD (550 nm) over ocean, QAExtend=3 for AE over land, QAExtend>=2 for AE over ocean, QAExtend=3 for SSA (550 nm) over land and ocean. Basically, the fulfilment of AOD GCOS requirement over land varies from ~48% (POSP and OLCI) to 57% (TROPOMI), over ocean, POSP AOD seems a bit less good than OLCI and TROPOMI, with GCOS fraction 54.3% for POSP relatively to 67.3% for OLCI and 71.3% for TROPOMI. For AE, all three products are qualitatively good to separate fine/coarse mode dominant cases, while quantitively there are clear tendency in over-/underestimation of AE for fine/coarse mode cases that need further refinement, particularly the particle size assumption for coarse mode in Models approach. For SSA, for OLCI validation, a larger AOD threshold (AOD>0.5) is used to select high-quality retrievals that lead to relatively small number of available points. TROPOMI shows slightly better performance that POSP in terms of RMSE (0.026 vs. 0.040) and fulfilment of Optimal requirement (+/- 0.03) (82.1% vs. 63.0%).

Overall, the performed AERONET validation results show that POSP/GF-5(02) GRASP aerosol product has good consistency with ground-based AERONET aerosol reference dataset, not only for AOD but also for detailed aerosol properties including AODF, AODC, AE as well as SSA. The validation results are basically comparable with previous aerosol products, such as OLCI/GRASP and TROPOMI/GRASP, in which TROPOMI product outperforms the other two products for most of statistical metrics. POSP AOD validation performance is comparable with OLCI, while AE over land

from POSP seems slightly better than OLCI, as well as the number of AE and SSA retrievals available for validation. This is a preliminary validation exercise for POSP aerosol product to obtain a general picture of the product quality and the instrument performance. In future, more in depth analysis of the product is certainly needed once more data is produced and the definition of the pixel-level quality flag is also in the to-do list for POSP product.

**4.3 Intercomparison of POSP, MODIS and VIIRS aerosol and surface product**

In this section, the POSP/GF-5(02) GRASP aerosol and surface product is intercompared with widely used NOAA and NASA aerosol and surface products, including NOAA-20 VIIRS DB aerosol product (AERDB_L2_VIIRS_NOAA20) and MCD43C3 surface white-sky-albedo. In order to compare the products at the same grid, we therefore re-grid POSP/GRASP 10 km, VIIRS/DB 6 km and MCD43C3 0.05° pixels onto common 0.2° x 0.2° grid box, and then the intercomparisons are performed for all 0.2° x 0.2° grid boxes globally. We still use similar criteria to select high quality retrievals for re-gridding.

Specifically, for POSP/GF-5(02) product, we require the relative fitting residual smaller than 5% over land and ocean; for VIIRS/DB, we use the original retrievals with quality assurance QA>=2 over land and ocean.

Figure 11 shows the global spatial distribution of GF-5(02) POSP/GRASP (Fig. 11a) and NOAA-20 VIIRS/DB (Fig. 11b) AOD (550 nm) in 2022, as well as the pixel level difference of daily AOD (550 nm) averaged over entire year (Fig. 11c).

On one hand, 2 products capture the global major aerosol features, such as Sahara dust, Taklimakan dust, southern Africa and southern America smoke, and high AOD over Indo-Gangetic Plain (IGP), etc. On the other hand, the 2 AOD products show non-neglectable differences in the AOD absolute values with spatial variability. For example, POSP/GRASP AOD (550 nm) is generally 0.05-0.15 higher than VIIRS/DB over desert region (Sahara, Taklimakan and Arabian Peninsula), and POSP/GRASP AOD (550 nm) is slightly 0.03-0.10 lower than VIIRS/DB over central and southern Africa biomass burning

region and India region. Statistically, the one-year global all grid boxes AOD (550 nm) metrics between POSP/GRASP and VIIRS/DB are presented in Figure 12. Based on ~10 million matchup grid boxes (5.3 million over land and 4.4 million over ocean), the grid-to-grid AOD (550 nm) linear correlation coefficients (R) between POSP/GRASP and VIIRS/DB are 0.675 over land and 0.746 over ocean, with differences (POSP-VIIRS) +0.043 ($1\sigma$=0.182) over land and -0.029 ($1\sigma$=0.074) over ocean. Generally, the AOD intercomparison results are reasonable while not as good as our previous exercise for

TROPOMI/Sentinel-5p vs. VIIRS/SNPP (Chen et al., 2024a). One intrinsic reason is the difference of satellite overpass time are few hours between GF-5(02) and NOAA-20 in contrast with few minutes between Sentinel-5p and SNPP. Besides, there are some potential reasons need to verify in the future, for example, the data quality, calibration accuracy, cloud mask, etc. Additionally, the ocean AOD (550 nm) validation in Fig. 6b shows that there seem some outliers in the POSP ocean products, for example the RMSE is 0.091 in contrast with typical values obtained from our previous processing of

TROPOMI and OLCI are around 0.04-0.05. Meanwhile, the intercomparison between POSP and VIIRS AOD (550 nm) over ocean also show larger STD 0.074 (Fig. 12) than 0.055 between OLCI and MODIS, and 0.033 between TROPOMI and VIIRS, indicating less stable in POSP AOD over ocean. There are three potential reasons: (i) the cloud mask doesn't

functional well and lead to cloud contamination in some pixels (Fig. 11), this can be an intrinsic reason due to its relative coarse spatial resolution (6.4 – 20 km) from POSP; (ii) the radiometric calibration issue in POSP SWIR channels; (iii) the wind speed is not used to constrain angular properties of ocean surface BRDF in POSP processing, which is proved to be able to stabilize the ocean AOD retrievals in previous OLCI and TROPOMI processing.

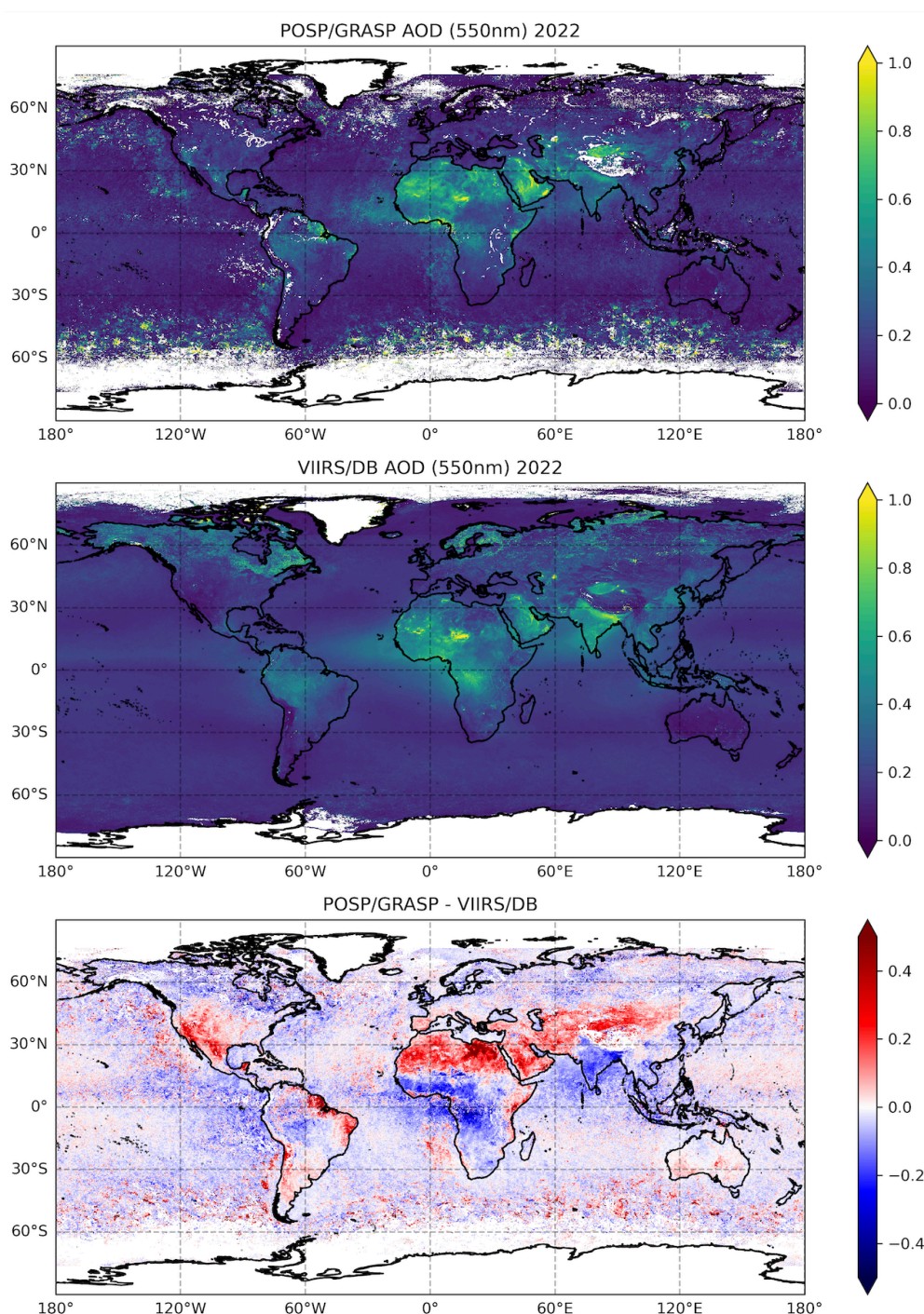

**Figure 11. Spatial distribution of one-year (2022) AOD (550 nm) from GF-5(02) POSP/GRASP and NOAA-20 VIIRS/DB. The pixel level difference of AOD 550 nm on daily basis between POSP and VIIRS averaged over a year is presented at the bottom as POSP/GRASP - VIIRS/DB.**

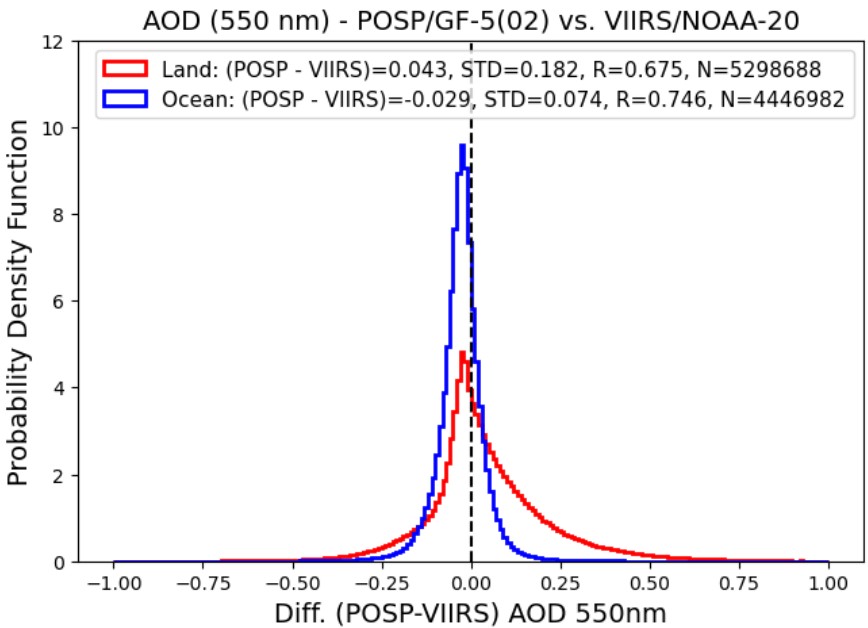

**Figure 12. Probability density function (PDF) of global all 0.2° x 0.2° grid box AOD (550 nm) differences between VIIRS/DB and POSP/GRASP over land and ocean.**

Figure 13 shows the global spatial distribution of GF-5(02) POSP/GRASP and NOAA-20 VIIRS/DB AODF (550 nm) and AODC (550 nm) in 2022, as well as the pixel level difference of daily AODF and AODC averaged over entire year. Note that the VIIRS/DB fine mode fraction (FMF) is only available over ocean provide by Satellite Ocean Aerosol Retrieval (SOAR) algorithm (Sayer et al., 2018a, 2018b). The intercomparison of AODF and AODC are performed over ocean only.

The one-year global all grid boxes AODF (550 nm) and AODC (550 nm) metrics between POSP/GRASP and VIIRS/DB are presented in Fig. 14. Some clear tendencies are identified for AODF and AODC over ocean. Specifically, POSP/GRASP AODF (550 nm) is almost everywhere 0.01 higher than VIIRS/DB, while POSP/GRASP AODC (550 nm) is generally 0.02-0.05 lower than VIIRS/DB. These results are also seen by AERONET validation that POSP seems to overestimate AODF while underestimate AODC over ocean (Figs. 8b and 9b), as well as the AE validation (Fig. 7b) that POSP seems to

underestimate particle size (overestimate AE) over ocean. This is potentially associate with the coarse models (dust and oceanic maritime) used in GRASP/Models. Overall, the agreement between 2 satellites AOD, AODF and AODC products generated with 2 independent algorithms are generally reasonable with R around 0.7-0.75, RMSE around 0.18 over land and 0.08 over ocean for total AOD, and RMSE around 0.05 for AODF and AODC over ocean.

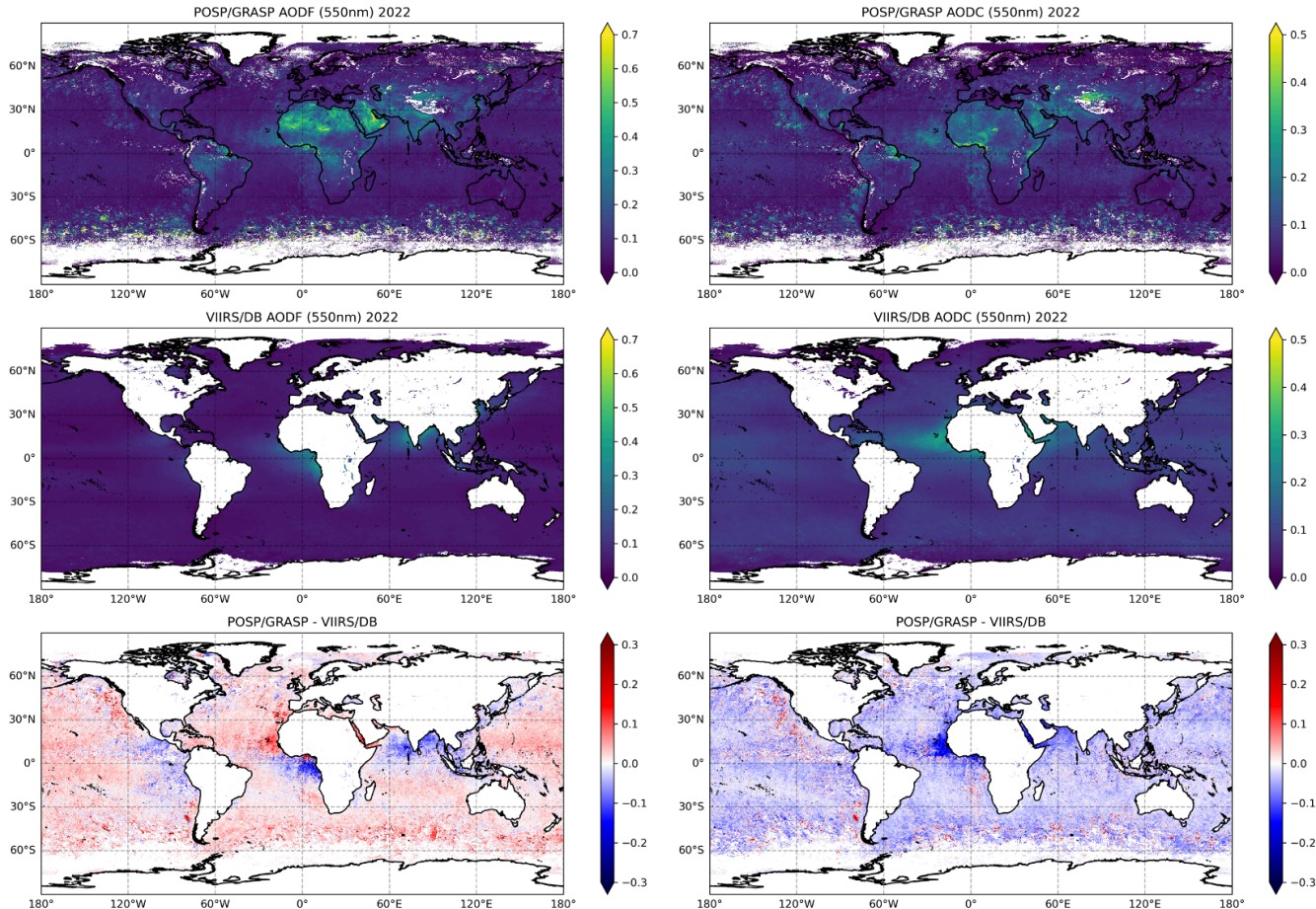

**Figure 13. Spatial distribution of one-year (2022) left column: AODF (550 nm) and right column: AODC (550 nm) from GF-5(02) POSP/GRASP and NOAA-20 VIIRS/DB. The pixel level difference of AODF and AODC at 550 nm on daily basis between POSP and VIIRS averaged over a year is presented at the bottom as POSP/GRASP - VIIRS/DB.**

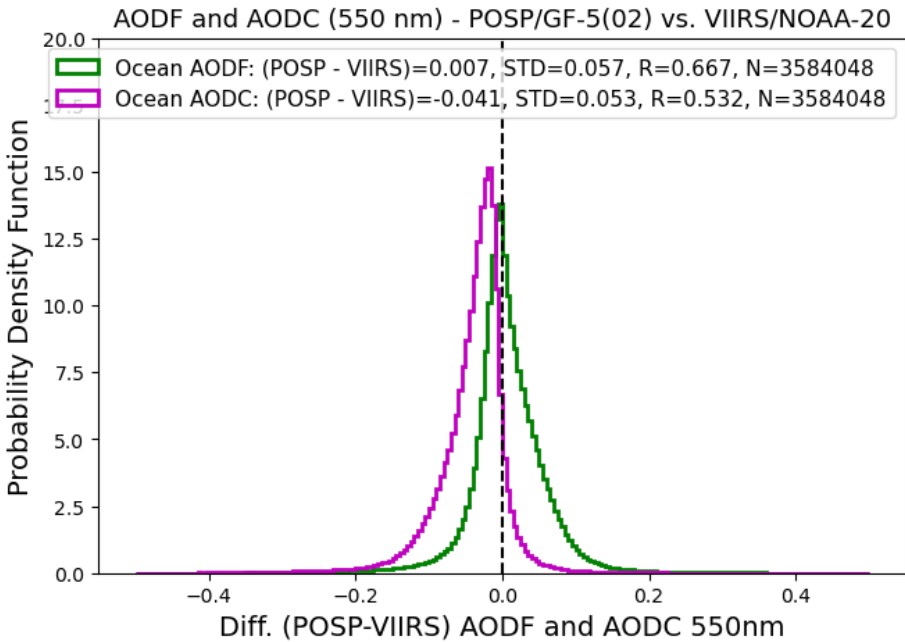

**Figure 14. Probability density function (PDF) of global all 0.2° x 0.2° grid box AODF (550 nm) and AODC (550 nm) differences between VIIRS/DB and POSP/GRASP over ocean.**

For surface product, we intercompare POSP/GF-5(02) GRASP and MODIS MCD43C3 surface white-sky-albedos at blue (442 nm vs. MODIS Band3), red (670 nm vs. MODIS Band1), NIR (865 nm vs. MODIS Band2) and SWIR (2254 nm vs. MODIS Band7) channels. The intercomparison strategy is the same as our previous study (Chen et al., 2022b, 2024a) that all global 0.2° x 0.2° grid box intercomparison is performed using monthly means. We should note that the MODIS MCD43C3 product is accumulation of 16-days TERRA and AQUA data and weighted to the day of interest, therefore we use the data

on the 15[th] day of each month to approximate the MODIS monthly mean surface albedos. In addition, the differences of the channel center wavelength and the bandwidth are different between POSP and MODIS. Therefore, the objective of the intercomparison exercise is to get a general idea of the 2 surface albedo products and analysis some trends that may be related to instrument performance.

Figure 15 shows the spatial distribution of one-year POSP/GF-5(02) and MODIS MCD43C3 BHRiso (white sky albedo) at blue, red, NIR and SWIR channels. Statistically, the one-year global all grid boxes BHRiso metrics between POSP/GRASP and MODIS MCD43 are presented in Fig. 16a, together with the monthly variation of the white-sky-albedo differences between POSP and MODIS in 2022 in Fig. 16b (visible channels) and Fig. 16c (NIR and SWIR channels). The surface Optimal (+/-0.01) and Target (+/-0.02) requirements used in this study are formulated in ESA S5P+I AOD/BRDF project

(Litvinov et al., 2024). Because surface is the dominant TOA signal in general, we observe quite consistent global spatial

variability between POSP and MODIS BHRiso at blue, red, NIR and SWIR channels. The best agreement between 2 products is observed at two visible channels (blue and red), especially at 670 nm. The global differences (POSP-MODIS) are around 0.002-0.003 with $1\sigma$ variation about 0.023, and the fulfillment of Target and Optimal requirements are ~75% and ~50% respectively. While, the differences become large at NIR (-0.022, $1\sigma$=0.038) and SWIR (-0.046, $1\sigma$=0.057) channels.

In addition, the SWIR channel difference is even larger (around -0.08 ~ -0.1) over Sahara and other desert regions, which is potentially connected with the overestimation of fine mode particles over desert regions (Fig. 4b and 4e). Certainly, we must acknowledge the difference of two channels (POSP 2254 nm vs. MODIS 2105-2155 nm). We should note the spectral difference can be quite significant in this range due to the absorption of different surfaces (Caplan and Huemmrich, 2025). Taking into account the new instrument POSP and the general challenges with the calibration of SWIR channels, we infer

that the calibration of POSP SWIR channels may underestimate high values. This is an inference from intercomparison analysis of indirect surface albedos that need further verification with Level 1 calibration team.

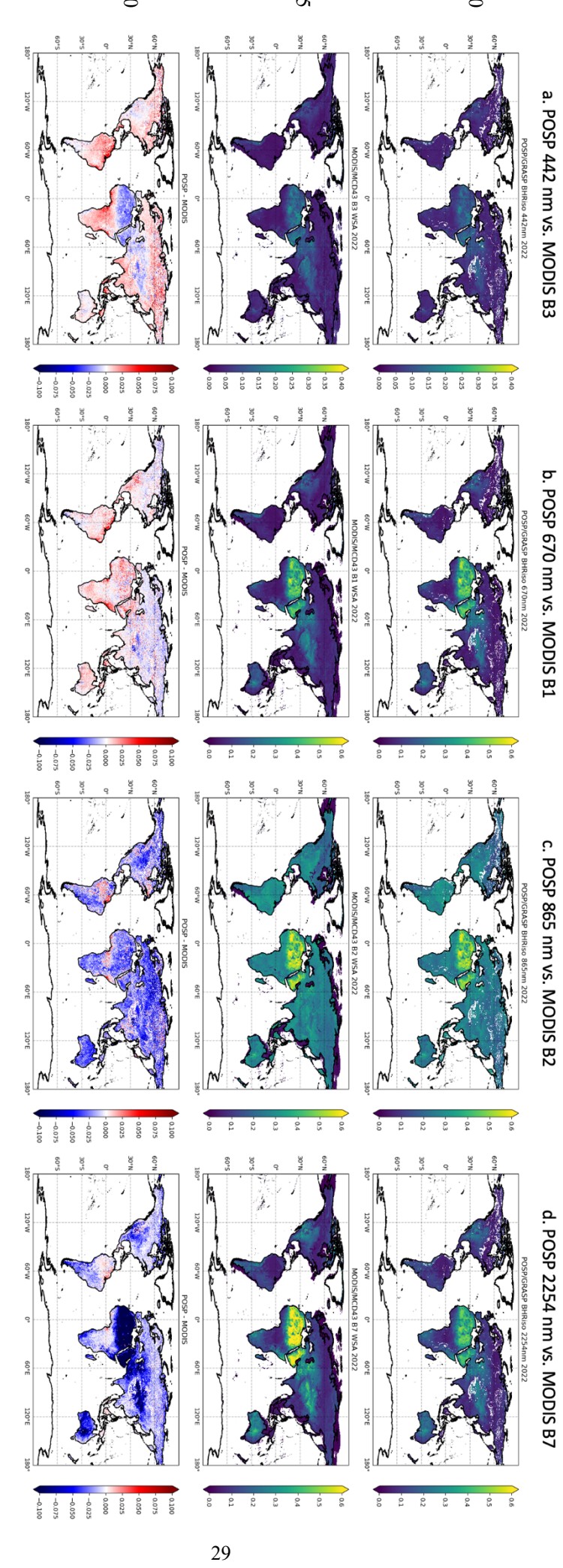

Figure 15. Spatial distribution of one-year (2022) POSP/GF-5(02) and MODIS MCD43 BHRiso (white sky albedo) at blue, red, NIR and SWIR channels. (a) POSP 442 nm vs. MODIS B3 459-479 nm; (b) POSP 670 nm vs. MODIS B1 620-670 nm; (c) POSP 865 nm vs. MODIS B2 841-876 nm; (d) POSP 2254 nm vs. MODS B7 2105-2155 nm. The pixel level difference on monthly basis between POSP and VIIRS averaged over a year is presented at the bottom as POSP/GRASP - VIIRS/DB.

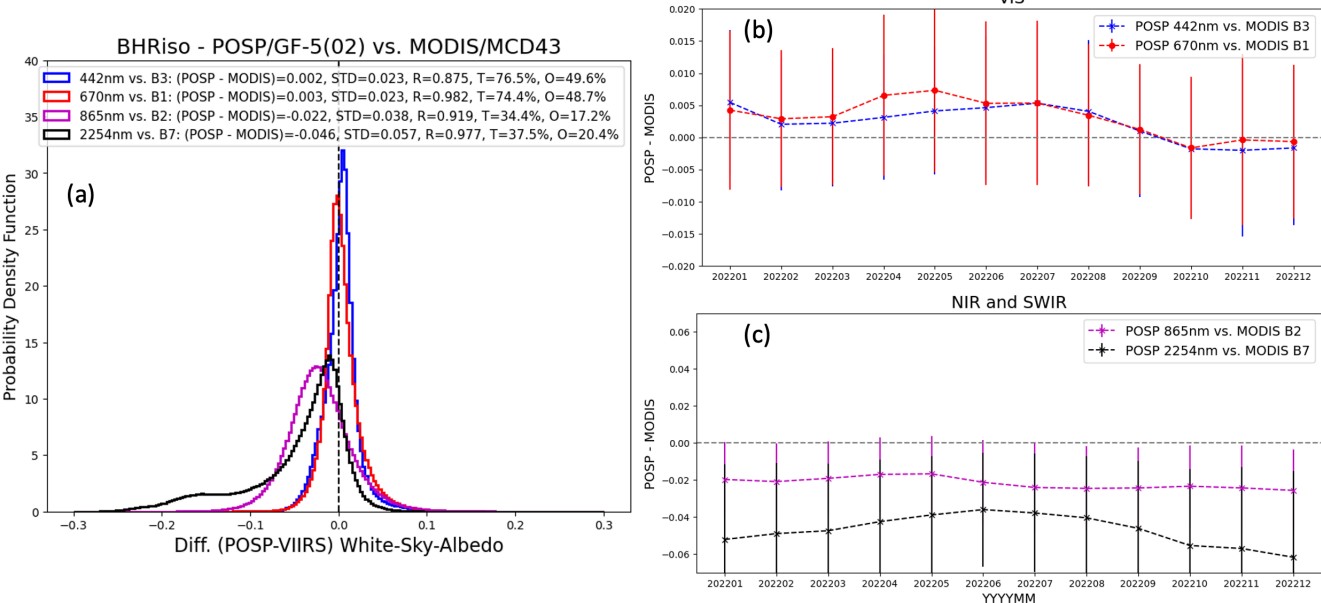

**Figure 16. (a) Probability density function (PDF) of global all 0.2° x 0.2° grid box white sky albedo differences between MODIS/MCD43 and POSP/GRASP. Monthly variation of the white-sky albedo differences between POSP and MODIS in 2022: (b) visible channels; (c) NIR and SWIR channels. The error bar indicates the 1σ variation of the differences (POSP-MODIS).**

## 5 Summary and conclusions

In this study, we provide a detailed description of the development of aerosol and surface Level 2 product and the processing scheme for the first space-borne UV-VIS-NIR-SWIR multi-spectral cross-track scanning polarimeter (POSP) onboard Chinese GF-5(02) satellite based on the GRASP/Models approach. Since POSP is a new polarimeter collecting both intensity and polarimetric measurements at UV, VIS, NIR and SWIR channels, we intend to verify the performance and feasibility of the instrument through the development of aerosol and surface products, validation and intercomparison with reference datasets and other independent satellite products. The total 18 months of POSP/GF-5(02) measurements from December 2021 to May 2023 were processed, and the obtained global aerosol and surface properties are evaluated and delivered as the baseline POSP product. We will continue to processing the data and deliver up-to date POSP/GF-5(02) aerosol and surface products.

Generally, we found the POSP single-viewing, multi-spectral (UV-VIS-NIR-SWIR) polarimetric measurements provide rich information content for aerosol and surface characterization, not only total aerosol optical depth, but also detailed properties

including aerosol size, absorption, layer height, type, etc. We provide spectral AOD, AODF, AODC, AE and SSA ranging from UV to SWIR, as well as aerosol scale height, columnar volume concentration for 4 aerosol models (BB – BC and OC/BC, Urban - sulfate, Oceanic – sea salt and dust) in the POSP/GF-5(02) aerosol product, and full BRDF, BPDF parameters, derived black-sky/white-sky albedos and vegetation index NDVI in the POSP/GF-5(02) surface product.

The validation of POSP one entire year (2022) aerosol product with ground-based AERONET reference data show general good agreement. Specifically, the correlation coefficients R for AOD (550 nm) are around 0.82-0.87 over land and ocean, and RMSEs are within 0.12 over land and ~0.09 over ocean. The fulfilment of AOD GCOS requirement is around 48% and 55% over land and ocean respectively. For aerosol particle size related parameters (AODF, AODC and AE), POSP AODF (550 nm) show good consistency with AERONET SDA product with R around 0.8 both over land and ocean, and bias is

within 0.015. POSP AODC (550 nm) has good correlation coefficient with AERONET that R is 0.93 over land and 0.79 over ocean, while POSP tends to overestimate (0.02) AODC over land and underestimate AODC (0.03-0.04) over ocean. Similarly, the validation of POSP AE indicates that POSP tends to underestimate particle size (overestimate AE) over ocean, and this tendency is also observed over bright land surface, especially over desert regions where POSP tends to overestimate AE and fine mode AOD.

The POSP/GF-5(02) aerosol and surface products are then intercompared with VIIRS/NOAA-20 DB aerosol and MODIS MCD43 surface products globally at common 0.2° x 0.2° grid box. Based on around 10 million grid box intercomparison, POSP/GRASP and VIIRS/DB total AOD (550 nm) show reasonable agreement with R around 0.7-0.75, RMSE around 0.18 over land and 0.08 over ocean, and POSP global mean AOD (550 nm) is about 0.04 higher than VIIRS over land and 0.03

smaller than VIIRS over ocean. Over land, POSP tends to report higher AOD than VIIRS over desert regions, where POSP seems to always overestimate fine mode contribution. This phenomenon can be associated with the POSP polarimetric calibration or SWIR channel radiometric calibration. For AODF and AODC over ocean, POSP/GRASP generally report higher AODF and lower AODC than VIIRS/DB. This also can be connected with POSP radiometric calibration and the coarse mode assumptions used in GRASP/Models approach. The intercomparison POSP/GRASP and MODIS MCD43

surface white-sky-albedos show overall good consistency in VIS channels (blue and red), while POSP tends to report smaller white-sky-albedos at NIR (-0.02) and SWIR (-0.045) channels than that from MODIS which is possibly due to the signal attenuation that require recalibration.

In the past, for Chinese satellite missions, ground segment is somehow insufficiently regarded as it should be. Here, some

605 lessons are learnt during the development of the global aerosol and surface products from POSP onboard GF-5(02) satellite. (i) Some useful and important pixel level identification or classification are missing or need to improve in Level 1 data. For

example, cloud mask is not well documented and functional well. We would recommend to include additional surface identifications, such as coastal line, mixed land and water, snow, ice, saline lake etc. and auxiliary data potential from reanalysis dataset, such as gas concentration, wind speed, etc., which are very helpful to develop high-quality Level 2 product. (ii) On-orbit calibration requires continuous efforts and consistent iteration with proper versioning of the product. Overall, the POSP/GF-5(02) baseline aerosol and surface Level 2 product is developed and will continue processing and release the product that verified the good performance of the first space-borne multi-spectral cross-track scanning polarimeter (POSP).

## Data availability

The developed POSP/GF-5(02) aerosol and surface products are publicly available and registered at doi.org/10.57760/sciencedb.14748 (Chen et al., 2024c).

## Author contribution

C. C., Z. Q. L. and J. H. convinced the initial idea for the study. X. L., Z. H. L., G. X., M. B. and Z. Q. support the Level 1 data preparation and Level 2 data processing resources. C. C., O. D., P. L., D. F. developed and supported the retrieval scheme. C. C., Z. Q. L., X. L. and H. G. performed the retrieval tests and carried out the processing. C.C. and Z. Q. L. wrote the manuscript with comments from all the authors.

## Competing interests

The authors declare that they have no conflict of interest.

## Acknowledgments

This research was supported by the National Key R&D Program of China (Grant No. 2024YF0811200), the National Outstanding Youth Foundation of China (Grant No. 41925019), the National Natural Science Foundation of China (Grant No. 42275144 and No. 42405130). We would like to thank the AERONET, MODIS and VIIRS teams for producing the dataset and making them available for the community.

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
