# Peer review of "Development of level 2 aerosol and surface products from cross-track scanning polarimeter POSP onboard GF-5(02) satellite"

_Earth System Science Data, 2024_

## Referee Comment (RC2)

This is a nice study which introduces and does some evaluation from a new aerosol data set derived from the POSP instrument on the GF-2 satellite. It is in scope to the journal and of interest to the readership. The contents are mostly what I would expect to find in a paper of this type.

The provided DOI works and the data are freely downloadable, which is great. I appreciate the "lessons learned" aspects of the discussion, both in terms of the GRASP algorithm and also issues related to e.g. ground segment and things like coast identification which are not always discussed.

That said, I have some questions about the work presented and the files themselves (which I opened in the Panoply tool to look at), and there are some important aspects of the data which are glossed over in the manuscript. I recommend revisions and would like to review the revised version. This in my view falls on the gap between minor and major revisions. My reasons for this recommendation are as follows:

1. The files are missing a lot of the important metadata which is commonly provided within satellite products from major agencies and institutions. For example there is no global metadata (e.g. originating institute, processing version info, contact etc). Variables have fill values, chunking, and coordinate systems specified but are missing a bunch of the other standard metadata e.g. long_name, valid_min, and valid_max. I strongly advise that these are added to the files for usability and operability issues. It should be possible via command line tools and scripts to add this metadata without having to reprocess the whole archive.

2. The unix time stamp looks strange and I am not sure how to interpret it. Could you check these values? I would expect it to increase monotonically during an orbit (from north to south or south to north dependent on direction), and between orbits (e.g. from east to west). Instead it jumps around a lot with artefacts near the middle of the orbit and some which seem out of place. If these patterns are correct then the orbit needs to be explained in more detail as I am not sure how a single satellite in a Sun-synchronous orbit could have overpass features like this on a single day. I have attached a screenshot below to show this.

[Figure]

unixtimestamp

unixtimestamp ()

1660198095.01660256423.81660314752.61660373081.41660431410.21660489739.0

Data Min = 1660198095.0, Max = 1660489739.0, Mean = 1660446133.9

3. The analysis is focused on validation at single locations (AERONET sites) and then global mapped time-composites of the year 2022. However, if you look at the data for an individual day, you notice a lot of strange things (see attached screenshots). For example, there are artefacts near the middle of the swaths in a lot of the variables which are clearly not physical. And the coverage of the different variables does not match exactly (some have fewer pixels than others). This is not documented in the files or discussed in the paper. Many users will be using the data on a daily basis so understanding these features is important to give an honest assessment of the data – particularly as there are no quality flags or uncertainty estimates provided in the files. This needs to be documented and discussed openly.

[Figure]

[Figure]

4. As a general comment Copernicus prefers not to use the rainbow color bar (because of the green in the middle) and suggests others such as viridis instead.

5. If I understand correctly, POSP is like APS but rotated so instead of collecting multiangle images along-track, it sees a wide across-track view (1850 km although the 10 outer pixels are stated to be skipped so I am not sure what the effective width is – could this be added?), but each location on the Earth is only seen from a single angle (so it's in effect a single-view, multi-spectral polarimeter). Is this right? If so I suggest expanding the text to write something like this as well, as otherwise people might see the text about APS and assume it is multi-angle too (because most polarimeters to now have been multiangle as well due to the added information content).

6. Lines 232-239: I am trying to figure out the multi-pixel configuration uses here as I know GRASP is flexible. I understand this is spatially, 3x3 pixels. Does the wording about NT also mean that all pixels from a given month are inverted simultaneously (i.e. the time period is 1 month)? What are the space/time

smoothness constraints for the retrieval as run? The paper should say what was used for this case.

7. Lines 293-301: This section describes the quality filtering applied for the validation analysis. The authors state that they did not put quality flags in the file. I suggest this is done, as it is not so practical for users to e.g. compute the 3x3 moving averages and counts everywhere and be confident about applying the residual threshold correctly. For example, there are two "*residual_relative_noise*" variables in the file and it is not clear which should be used for this test. Otherwise, the data as presented will not be consistent with the data filtering used for the analyses in the paper.

8. Figure 6: Almost all the points are below AOD of 0.2 which is buried in one corner of the plot because AOD data are highly skewed. I suggest showing this on a log scale (maybe truncate at 0.01 on the lower end) as this would show the magnitude and direction of any biases more directly. I also think the fit line would make more sense shown on log scale for this same reason about distribution shape.

9. Figures 6-10: Could you explain the color shading here? I initially thought it was density of points (i.e. a heat map aka scatter density plot). But, looking more closely the data are shown as a scatter plot instead of a heat map. And there is no color bar on the figures. If this is a heat map, then it should be shown with solid boxes and a color bar. If it is a scatter plot, then showing colors is just confusing. It implies the data are clustered in a certain way by drawing the eye, but it is not documented in the paper as far as I can tell what it means. My preference would be for a heat map because the meaning is clear and more informative than just a scatter plot.

10. Figure 11: is this (1) a difference in the mean (i.e. top panel minus bottom panel) or (2) the mean of the differences calculated on a daily basis? This is not clear and should be stated. In my opinion option (2) is better because it decreases sampling-related differences by ensuring both instruments saw the same location on the same day. And if there is a concern (e.g. line 395) about overpass times causing a difference due to e.g. aerosol transport, this would also be a smaller uncertainty source if the comparison were done at a coarser spatial scale (e.g. 0.5 or 1 degree instead of 0.2). (Note, I did not see the GF satellite orbit times listed in the paper, from the swath patterns I guess it is descending during the daytime node, what is the Equatorial crossing time?)

11. Figures 12, 14, 16: Should be "Probability" density function not "Possibility" in the caption.

12. Figures 13,15: same question/suggestion as Figure 11.

---

## Referee Comment (RC3)

This manuscript developed new aerosol and surface products from the first 18 months multi-spectral and polarized measurements of POSP/GF-5(02) based on the Generalized Retrieval of Atmosphere and Surface Properties (GRASP)/Models approach. These products are validated and intercompared with ground-based aerosol inversion dataset and other independent satellite aerosol and surface products. The results show generally good consistency of POSP products including not only total Aerosol Optical Depth (AOD), but also detailed aerosol properties such as aerosol size, absorption, layer height, type, etc., as well as full surface Bidirectional Reflectance Distribution Function (BRDF), Bidirectional Polarization Distribution Function (BPDF), black-sky, white-sky albedos and Normalized Difference Vegetation Index (NDVI). This research deserves to being published given the new valuable satellite products development, but there are still some descriptions and statements unclear and need to be improved. The detailed comments can be found below.

1. Introduction: I think most parts of the first paragraph needs to be re-written. POSP is a single-viewing multi-spectral polarimetric sensor aimed at aerosol detection, so I suppose this paragraph should introduce the research background about using multi-spectral or polarized measurements to retrieve aerosol properties. However, the authors mainly discuss aerosol retrievals from multi-angle polarimetric measurements, while multi-angle is not one of the characteristics of POSP. I suggest to focus on the characteristics of POSP and introduce the underlying fundamental physics here.

2. Section 2.1: It is mentioned that "POSP is the first space-borne multi-spectral cross-track scanning polarimeter". Why does POSP use cross-track scanning method instead of along-track scanning? What are the advantages and disadvantages of cross-track scanning? Some background can be added here.

3. Table 1: What is POSP/GF5 overpass time? I suggest to add it in this table and compare it with NOAA-20 since the difference of their overpass time can cause inconsistency in aerosol retrievals as mentioned in the analysis later.

4. Line 203-221: Although these land/ocean surface models including BRDF and BPDF have been discussed in many previous studies, I think some statements are still needed to clarify the meaning of related parameters involved in the state vector, such as $a_{iso}$, $a_{vol}$ and $a_{geom}$ (I assume they are the linear coefficients of three kernels in Ross-Li BRDF model), as well as $r_0$, $\delta_{Fr}$ and $\sigma^2$. Is BPDF only considered for land surface but not ocean?

5. Figure 4: It seems the AE and scale height retrieval are less than other parameters, shown as many blank pixels over ocean in (e) and (f). What are the reasons for this situation? Are there any different criteria applied for different parameters retrieval availability or quality?

6. In the scatter plots, such as Figure 6-10, what does the color of each dot mean? This should be added in the caption.

---

## Author Comment (AC1)

We would like to thank the three referees and the editor for their time reviewing the manuscript, and for the helpful feedback provided. The detailed responses to all referees are provided below.

**Reviewer #1:**

The study is well-organized, and the results demonstrate good agreement with other datasets. However, several concerns need to be addressed, and providing additional information will further strengthen the paper.

**Response:**

We would like to thank the reviewer for your time reviewing the manuscript and appreciate the constructive comments on our paper.

1. Are the images in Figures 4 and 5 averaged over the entire 18-month period? While it is clear that data was processed and made publicly available for 18 months, averaging all 18 months together may lead to an unbalanced seasonal representation, where certain seasons dominate the dataset. This results in an image that is neither an annual mean nor a seasonal mean, making it difficult to interpret. Additionally, the AERONET validation later in the paper appears to use only one year of data. This raises the question of whether it is necessary to average all 18 months in these figures. Also, I think it would be better to have more descriptions for Figure 4 and 5

**Response:**

Yes, the images in Figures 4 and 5 are averaged over the entire 18-months of processed POSP data. Basically, Figures 4 and 5 are illustrations of the main aerosol and surface products provided in the POSP/GRASP level 2 product. The spatial distribution of averaged 18-months of data is to keep consistency with Figure 3, in which the total number of valid retrievals over the entire 18-month period is present. The 2022 annual

mean of the main POSP/GRASP aerosol and surface products are present in Figures A1 and A2. Note, the annual mean map of POSP AOD, AODF, AODC at 550 nm and BHRiso at 442, 670, 865 and 2254 nm are also shown in Figures 11, 13 and 15. Therefore, we would keep the Figures 4 and 5 as an illustration of the main aerosol and surface products. We have added more descriptions to Figures 4 and 5.

[Figure]

Figure A1. Spatial distribution of POSP/GF-5(02) 2022 annual mean main aerosol products: (a) Aerosol Optical Depth – AOD; (b) Fine mode Aerosol Optical Depth – AODF; (c) Coarse mode Aerosol Optical Depth (AODC); (d) Single Scattering Albedo – SSA; (e) Ångström Exponent – AE (440/870); (f) Scale height of aerosol vertical profile – ALH; Note AOD, AODF, AODC, SSA are spectral dependent and provided at UV, VIS, NIR and SWIR spectrum.

[Figure]

Figure A2. Spatial distribution of POSP/GF-5(02) 2022 annual mean main surface products: (a) Ross Li BRDF isotropic parameter; (b) Ross Li BRDF normalized volumetric parameter; (c) Ross Li BRDF normalized geometric parameter; (d) Maignan-Bréon BPDF; (e) Surface Isotropic Bihemispherical Reflectance – BHRiso or White Sky Albedo; (f) Normalized Difference Vegetation Index – NDVI. Note BRDF1 and BHRiso (White Sky Albedo) are spectral dependent and provided at UV, VIS, NIR and SWIR spectrum.

2. In line 293, it is mentioned that SSA data was matched within ±180 minutes, which is different from the other validation datasets. It would be helpful to explain why this specific time window was chosen.

**Response:**

Thanks for the suggestion! In this study, 3 types of AERONET Version 3 Level 2 aerosol products are used to verify the results obtained from POSP/GRASP processing, including

direct-Sun, SDA and Inversion products. The AERONET CIMEL sun-photometer carries out direct-Sun measurements about every 15 mins and generate the direct-Sun spectral AOD, AE products and derived SDA fine/coarse mode AOD products. While, the AERONET inversion products are generated based on the sky-scanning Almucantar measurements which are carried out hourly from 9:00 to 15:00 local time or optical air mass equals to 4, 3, 2, 1.7 both in the morning and the afternoon. Therefore, the inversion products of SSA, complex refractive index, particle size distribution are provided less frequent than the direct-Sun measurements of AOD, AE, AODF and AODC. In order to collocated satellite overpass time and AERONET products, we use +/- 30 mins for AOD, AE, AODF and AODC validation and +/- 180 mins for SSA validation. Also, several studies (Sayer, 2020; Schutgens et al., 2016) suggest that aerosol type related parameter, such as SSA, vary temporally slowly than that aerosol concentration related parameters, such as AOD.

Schutgens, N. A. J., Partridge, D. G., and Stier, P.: The importance of temporal collocation for the evaluation of aerosol models with observations, *Atmos. Chem. Phys.*, 16, 1065–1079, https://doi.org/10.5194/acp-16-1065-2016, 2016.

Sayer, A. M. (2020). How long is too long? Variogram analysis of AERONET data to aid aerosol validation and intercomparison studies. *Earth and Space Science*, 7(9), e2020EA001290.

3. In lines 316–318 and 369–370, the paper states that the results are similar to previous studies. However, instead of just stating that they are similar, it would be more effective to include specific numerical comparisons in the form of figures or tables to support this claim.

**Response:**

Thanks for the suggestion! We have added a table (Table A1) to list some key statistical parameters for one-year AERONET validation of AOD, AE, and SSA obtained from POSP, OLCI and TROPOMI for intercomparison. Basically, in most of statistical metrics, TROPOMI product outperforms the other two products. POSP AOD validation performance is comparable with OLCI, while AE over land and SSA from POSP seems slightly better than OLCI.

Table A1. Intercomparisons of OLCI/Sentinel-3A, TROPOMI/Sentinel-5p and POSP/GF-5(02) one-year AOD (550 nm), AE (OLCI and POSP: 440/870; TROPOMI: 412/670) and SSA (550 nm) one-year validation metrics with AERONET reference dataset. The best performance metric is indicated in bold.

| | Sensor (num. pairs) | R | RMSE | BIAS | Optimal% | Target% |
|---|---|---|---|---|---|---|
| AOD 550 nm (Land) | OLCI (3205) | 0.870 | 0.090 | -0.01 | 47.9 | - |
| | TROPOMI (**7732**) | **0.885** | **0.078** | 0.01 | **56.9** | **67.9** |
| | POSP (1667) | 0.813 | 0.114 | **0.004** | 47.6 | 57.0 |
| AOD 550 nm (Ocean) | OLCI (217) | 0.872 | **0.047** | **0.01** | 67.3 | - |
| | TROPOMI (**880**) | **0.881** | **0.047** | **0.01** | 71.3 | 81.1 |
| | POSP (219) | 0.861 | 0.091 | -0.012 | 54.3 | 66.7 |
| AE (Land) | OLCI (611) | 0.526 | 0.531 | -0.35 | - | - |
| | TROPOMI (**991**) | **0.725** | **0.418** | **0.18** | 51.6 | 76.0 |
| | POSP (719) | 0.270 | 0.472 | - | 49.4 | 73.3 |
| AE (Ocean) | OLCI (46) | **0.751** | 0.386 | 0.18 | - | - |
| | TROPOMI (**1417**) | 0.474 | 0.474 | **0.16** | 52.1 | 71.3 |
| | POSP (158) | 0.409 | 0.667 | - | 29.7 | 49.4 |
| SSA 550 nm (Land + Ocean) | OLCI (115) | 0.471 | **0.026** | -0.01 | - | - |
| | TROPOMI (358) | **0.522** | **0.026** | **0.00** | 82.1 | 94.4 |
| | POSP (**570**) | 0.305 | 0.040 | -0.007 | 63.0 | 81.2 |

4. In Figure 6, while it is true that the ocean validation results appear better than those for land and that most data points are well distributed around the 1:1 line, there seem to be several significant outliers. In other studies, ocean validation results have generally been more stable. Of course, as the number of validation points increases,

some outliers are inevitable, but considering that Figure 6 shows only 219 ocean data points, the distribution appears somewhat scattered. Since the paper already mentions that the glint mask was applied too strictly and should be improved, this is likely not a glint-related issue. Instead, it would be helpful to provide additional explanations on whether the issue is due to bad pixel screening, a limitation in GRASP's ocean retrieval algorithm, or some other factor.

**Response:**

Thanks for the suggestion! I fully agree that the ocean validation in Figure 6(b) shows that there seem some outliers in the POSP ocean products, for example the RMSE is 0.091 in contrast with typical values obtained from our previous processing of TROPOMI and OLCI are around 0.04-0.05. Meanwhile, the intercomparison between POSP and VIIRS AOD (550 nm) over ocean also show larger RMSE 0.074 (Figure 12) than 0.055 between OLCI and MODIS, and 0.033 between TROPOMI and VIIRS, indicating less stable in POSP AOD over ocean. There are three potential reasons: (i) the cloud mask doesn't functional well and lead to cloud contamination in some pixels (Figure 11), this can be an intrinsic reason due to its relative coarse spatial resolution (6.4 – 20 km) from POSP; (ii) the radiometric calibration issue in POSP SWIR channels; (iii) the wind speed is not used to constrain angular properties of ocean surface BRDF in POSP processing, which is proved to be able to stabilize the ocean AOD retrievals in previous OLCI and TROPOMI processing. We have added these to the discussions in the main text.

---

## Author Comment (AC2)

We would like to thank the three referees and the editor for their time reviewing the manuscript, and for the helpful feedback provided. The detailed responses to all referees are provided below.

**Reviewer #2:**

This is a nice study which introduces and does some evaluation from a new aerosol data set derived from the POSP instrument on the GF-5(02) satellite. It is in scope to the journal and of interest to the readership. The contents are mostly what I would expect to find in a paper of this type.  The provided DOI works and the data are freely downloadable, which is great. I appreciate the "lessons learned" aspects of the discussion, both in terms of the GRASP algorithm and also issues related to e.g. ground segment and things like coast identification which are not always discussed. That said, I have some questions about the work presented and the files themselves (which I opened in the Panoply tool to look at), and there are some important aspects of the data which are glossed over in the manuscript. I recommend revisions and would like to review the revised version. This in my view falls on the gap between minor and major revisions. My reasons for this recommendation are as follows:

**Response:**

We would like to thank the reviewer for your time reviewing the manuscript and appreciate the constructive comments on our paper.

1. The files are missing a lot of the important metadata which is commonly provided within satellite products from major agencies and institutions. For example there is no global metadata (e.g. originating institute, processing version info, contact etc). Variables have fill values, chunking, and coordinate systems specified but are missing a bunch of the other standard metadata e.g. long_name, valid_min, and valid_max. I

strongly advise that these are added to the files for usability and operability issues. It should be possible via command line tools and scripts to add this metadata without having to reprocess the whole archive.

**Response:**

Thanks very much for the suggestion! We have revised all Level 2 files and update a new version (v3.0) under the registered doi.org/10.57760/sciencedb.14748. Meanwhile, we have added detailed descriptions about the dataset to the webpage, including product specification table, selection of high-quality retrievals criteria, as well as the known issues.

2. The unix time stamp looks strange and I am not sure how to interpret it. Could you check these values? I would expect it to increase monotonically during an orbit (from north to south or south to north dependent on direction), and between orbits (e.g. from east to west). Instead it jumps around a lot with artefacts near the middle of the orbit and some which seem out of place. If these patterns are correct then the orbit needs to be explained in more detail as I am not sure how a single satellite in a Sun-synchronous orbit could have overpass features like this on a single day. I have attached a screenshot below to show this.

[Figure]

unixtimestamp

unixtimestamp ()

1660198095.01660256423.81660314752.61660373081.41660431410.21660489739.0

Data Min = 1660198095.0, Max = 1660489739.0, Mean = 1660446133.9

**Response:**

Thanks for careful checking the data files! Indeed, there are some issues in the Level 2 file merging. Because the temporary files in the processing are spliced into land/ocean and then merged them into single Level 2 netcdf files. The merged Level 2 is not consistent with original Level 1C prepared for processing (see Figure A3). We have revised the field in the Level 2 files.

[Figure]

Figure A3. The discrepancy of the field UNIX timestamp between Level 1C and Level 2 files.

3. The analysis is focused on validation at single locations (AERONET sites) and then global mapped time-composites of the year 2022. However, if you look at the data for an individual day, you notice a lot of strange things (see attached screenshots). For example, there are artefacts near the middle of the swaths in a lot of the variables which are clearly not physical. And the coverage of the different variables does not match exactly (some have fewer pixels than others). This is not documented in the files or discussed in the paper. Many users will be using the data on a daily basis so

understanding these features is important to give an honest assessment of the data –
particularly as there are no quality flags or uncertainty estimates provided in the files.
This needs to be documented and discussed openly.

[Figure]

**Response:**

Thanks very much for the suggestion! We investigate and figure out there is an issue in
the Level 0 to Level 1 data preparation that cause these strange lines in the middle of the
swath. Since there may be a delay in the production of recorded Level 0 data to Level 1
data, some L1 tracks are recorded until the next day, resulting in stripes in the L1 data
itself (see an example in Figure A4). This is a fundamental problem in our entire
processing chain, and to fully resolve it we will have to reprocess the entire data set. At
this stage, we have listed the possible problematic tracks (236 orbits in total) and dates in
the Known Issues with the dataset (Table A2). Meanwhile, we also estimate the area with
overlaps with six points describing the rectangle area, and the coordinates are also

provided in the dataset description webpage. Overall, this is the baseline processing of POSP data. Our main goal is to find out issues and continuously improve them in subsequent processing.

We have added a short paragraph to describe the known issues.

"During the first POSP/GF-5(02) processing, we also identify some remain issues in the current baseline Level 2 products. (i) The cloud and glint mask over ocean seems too strict resulting in the ocean pixels percentage is much lower than expected; (ii) We also identify some existing stripes in the Level 2 aerosol and surface products that are caused by the delay of Level 1 data production, therefore some Level 1 tracks are recorded until the next day and overwrite the coming data. These known issues are documented in the data description and expected to be solved in the next processing.".

[Figure]

Figure A4. An example shows the delay in generating POSP level 0 to level 1 data, some level 1 data is overwritten and result in stripes in L2 aerosol and surface products.

Table A2. List of the orbit numbers with data overwritten issue.

| Date | orbit number | Date | orbit number | Date | orbit number | Date | orbit number |
|---|---|---|---|---|---|---|---|
| 20211206 | 1279 | 20220618 | 4094 | 20221008 | 5633 | 20230202 | [7427;7431] |
| 20211213 | [1367;1368;1369] | 20220623 | 4152 | 20221013 | 5785 | 20230206 | 7444 |
| 20211216 | [1383;1384] | 20220627 | [4221;4222;4226] | 20221015 | [5805;5806] | 20230214 | 7598 |
| 20211218 | 1428 | 20220701 | 4284 | 20221017 | 5829 | 20230224 | 7736 |
| 20211222 | 1487 | 20220703 | [4295;4296] | 20221021 | 5900 | 20230226 | 7752 |
| 20220105 | 1711 | 20220710 | [4394;4395] | 20221023 | [5943;5944] | 20230304 | 7853 |
| 20220214 | [2278;2279;2280] | 20220712 | [4422;4425;4426;4429] | 20221025 | 5974 | 20230308 | 7910 |
| 20220215 | 2318 | 20220716 | [4501;4502] | 20221028 | 5973 | 20230314 | 8002 |
| 20220216 | [2314;2315;2316] | 20220724 | 4622 | 20221030 | [6043;6045] | 20230316 | 8037 |
| 20220217 | 2325 | 20220725 | [4643;4644] | 20221101 | 6060 | 20230318 | 8041 |
| 20220219 | 2281 | 20220728 | [4650;4651;4653;4658;4659] | 20221102 | [6089;6090] | 20230330 | 8234 |
| 20220221 | [2372;2373] | 20220731 | [4720;4721;4722] | 20221105 | 6108 | 20230405 | 8334 |
| 20220315 | [2691;2692] | 20220801 | [4734;4735] | 20221107 | [6147;6148] | 20230409 | [8373;8374] |
| 20220317 | 2730 | 20220802 | 4747 | 20221108 | [6177;6178;6179] | 20230416 | [8492;8493] |
| 20220319 | [2776;2777;2778;2779;2780] | 20220804 | 4763 | 20221109 | 5292 | 20230418 | 8523 |
| 20220323 | 2783 | 20220806 | 4786 | 20221110 | 6173 | 20230420 | 8541 |
| 20220324 | 2823 | 20220808 | [4812;4814] | 20221112 | 6221 | 20230425 | 8622 |
| 20220328 | [2896;2897;2898] | 20220814 | [4921;4923;4925;4926] | 20221114 | [6249;6250;6251] | 20230427 | 8635 |
| 20220330 | [2922;2923] | 20220815 | 4922 | 20221116 | [6278;6279;6280] | 20230501 | 8695 |
| 20220401 | [2950;2953] | 20220816 | [4928;4929;4933;4937;4938;4941] | 20221118 | 6146 | 20230504 | 8755 |
| 20220406 | [3026;3027] | 20220817 | [4930;4934;4936;4939] | 20221124 | [6422;6423] | 20230517 | [8945;8946;8947;8949;8950] |
| 20220408 | 3044 | 20220819 | 4927 | 20221126 | 6441 | 20230519 | 8958 |
| 20220416 | 3171 | 20220821 | 4932 | 20221128 | 6437 | 20230523 | [9014;9015] |
| 20220419 | 3172 | 20220822 | 4932 | 20221130 | 6499 | | |
| 20220422 | [3275;3276;3277] | 20220823 | [5039;5041] | 20221202 | 6510 | | |
| 20220425 | [2691;2692] | 20220825 | 5075 | 20221204 | 5291 | | |
| 20220427 | 3320 | 20220828 | [5111;5112] | 20221209 | 6645 | | |
| 20220505 | 3439 | 20220901 | [5178;5180] | 20221212 | [6683;6684] | | |
| 20220513 | [3579;3580] | 20220907 | [5251;5256;5257] | 20221221 | [6814;6815] | | |
| 20220516 | [3625;3626] | 20220909 | [5284;5285] | 20221226 | [6871;6876;6878] | | |
| 20220517 | 3639 | 20220913 | 5314 | 20221228 | 6862 | | |
| 20220519 | 3624 | 20220915 | [5369;5371] | 20221230 | 6891 | | |
| 20220523 | [3714;3715] | 20220919 | 5450 | 20230101 | 6964 | | |
| 20220524 | 3688 | 20220920 | 5366 | 20230103 | [6968;6969] | | |
| 20220527 | [3783;3784;3785] | 20220923 | [5506;5507] | 20230109 | 7094 | | |
| 20220602 | 3835 | 20220925 | [5535;5536] | 20230110 | 7104 | | |
| 20220606 | [3915;3917] | 20220926 | 5538 | 20230111 | [7106;7109] | | |
| 20220612 | 3978 | 20220927 | 5546 | 20230119 | 7215 | | |
| 20220613 | 4023 | 20220928 | 5517 | 20230123 | 7271 | | |
| 20220616 | 4069 | 20220929 | 5545 | 20230129 | 7358 | | |
| | | 20221007 | 5711 | 20230131 | 7275 | | |

4. As a general comment Copernicus prefers not to use the rainbow color bar (because of the green in the middle) and suggests others such as viridis instead.

**Response:**

Thanks! It's good to know. We have revised all figures to use "viridis" color bar.

5. If I understand correctly, POSP is like APS but rotated so instead of collecting multiangle images along-track, it sees a wide across-track view (1850 km although the 10 outer pixels are stated to be skipped so I am not sure what the effective width is – could this be added?), but each location on the Earth is only seen from a single angle (so it's in effect a single-view, multi-spectral polarimeter). Is this right? If so I suggest expanding the text to write something like this as well, as otherwise people might see the text about APS and assume it is multi-angle too (because most polarimeters to now have been multiangle as well due to the added information content).

**Response:**

Yes, POSP is a cross-track single-viewing polarimeter. The POSP cross-track field of view (FOV) is about +/- 64°, with an angular interval of 0.52° per sampling point and 224 ground pixels. After removing 10 sampling points from each end, 204 points remain, corresponding to a relative off-nadir angle of +/-53°. Accounting for Earth's curvature, this results in a ground swath width of 2,110 km, with the spatial resolution at the edge reaching 25 km. The specified engineering requirement for POSP is +/-50° off-nadir angle, corresponding to a swath width of 1,850 km. Thanks for the suggestion! We have added this information to the main text and updated Table 1.

6. Lines 232-239: I am trying to figure out the multi-pixel configuration uses here as I know GRASP is flexible. I understand this is spatially, 3x3 pixels. Does the wording about NT also mean that all pixels from a given month are inverted simultaneously (i.e. the time period is 1 month)? What are the space/time used for this case.

**Response:**

Yes, we inverted temporal 1 month (NT) of spatially 3x3 pixels simultaneously. Basically, all available pixels within the 3x3xNT segment are retrieved simultaneously.

7. Lines 293-301: This section describes the quality filtering applied for the validation analysis. The authors state that they did not put quality flags in the file. I suggest this is done, as it is not so practical for users to e.g. compute the 3x3 moving averages and counts everywhere and be confident about applying the residual threshold correctly. For example, there are two "*residual_relative_noise*" variables in the file and it is not clear which should be used for this test. Otherwise, the data as presented will not be consistent with the data filtering used for the analyses in the paper.

**Response:**

Thanks for the suggestion! In the netcdf file *residual_relative_noise0* represents the relative fitting residual of measered reflectance, and *residual_relative_noise1* represents the relative fitting residual of measered degree of polarization. In the validation part, we use *residual_relative_noise0* to make quality filtering. This information has been added to Table 2.

8. Figure 6: Almost all the points are below AOD of 0.2 which is buried in one corner of the plot because AOD data are highly skewed. I suggest showing this on a log scale (maybe truncate at 0.01 on the lower end) as this would show the magnitude and direction of any biases more directly. I also think the fit line would make more sense shown on log scale for this same reason about distribution shape.

**Response:**

Thanks for the suggestion! Agree, we have added the AOD, AODF and AODC validation figures with log-log scale together with the original linear scale plots (see an example in Figure A5).

[Figure]

Figure A5. Validation of POSP/GF-5(02) GRASP AOD (550 nm) with AERONET over land and ocean for an entire year 2022. (a) POSP AOD validation with linear scale over land; (b) POSP AOD validation with logarithmic scale over land; (c) POSP AOD validation with linear scale over ocean; (d) POSP AOD validation with logarithmic scale over ocean.

9. Figures 6-10: Could you explain the color shading here? I initially thought it was density of points (i.e. a heat map aka scatter density plot). But, looking more closely the data are shown as a scatter plot instead of a heat map. And there is no color bar on the figures. If this is a heat map, then it should be shown with solid boxes and a color bar. If it is a scatter plot, then showing colors is just confusing. It implies the data are clustered in a certain way by drawing the eye, but it is not documented in the paper as far as I can

tell what it means. My preference would be for a heat map because the meaning is clear and more informative than just a scatter plot.

Response:

Yes, Figures 6-10 are scatter plots and the color represents the probability density function (PDF) of data pairs (x, y). We use kernel density estimation to calculate the PDF of data pairs (x, y). In revised manuscript, we replaced the scatter plot to heat map, for example in Figure A5, which is coinvent to interpret that the color represents the number of valid points in each 0.01x0.01 grid. While, we keep the scatter plot for the logarithmic scale plots.

10. Figure 11: is this (1) a difference in the mean (i.e. top panel minus bottom panel) or (2) the mean of the differences calculated on a daily basis? This is not clear and should be stated. In my opinion option (2) is better because it decreases sampling-related differences by ensuring both instruments saw the same location on the same day. And if there is a concern (e.g. line 395) about overpass times causing a difference due to e.g. aerosol transport, this would also be a smaller uncertainty source if the comparison were done at a coarser spatial scale (e.g. 0.5 or 1 degree instead of 0.2). (Note, I did not see the GF satellite orbit times listed in the paper, from the swath patterns I guess it is descending during the daytime node, what is the Equatorial crossing time?)

Response:

Yes, the mean of difference in Figures 11 and 13 are calculated on a daily basis. GF-5(02) satellite is in the descending node during the daytime and equatorial crossing time is 10:30 local time. We have added this information to Table 1.

11. Figures 12, 14, 16: Should be "Probability" density function not "Possibility" in the caption.

**Response:**

Revised.

12. Figures 13,15: same question/suggestion as Figure 11.

**Response:**

The mean of difference in Figure 13 for AODF and AODC are calculated on a daily basis. While it's calculated on a monthly basis for surface parameters in Figure 15, since the surface properties are temporal stable and MODIS MCD43 product is obtained by accumulating 16-days TERRA and AQUA data and weighted to the day of interest.

---

## Author Comment (AC3)

We would like to thank the three referees and the editor for their time reviewing the manuscript, and for the helpful feedback provided. The detailed responses to all referees are provided below.

**Reviewer #3:**

This manuscript developed new aerosol and surface products from the first 18 months multispectral and polarized measurements of POSP/GF-5(02) based on the Generalized Retrieval of Atmosphere and Surface Properties (GRASP)/Models approach. These products are validated and intercompared with ground-based aerosol inversion dataset and other independent satellite aerosol and surface products. The results show generally good consistency of POSP products including not only total Aerosol Optical Depth (AOD), but also detailed aerosol properties such as aerosol size, absorption, layer height, type, etc., as well as full surface Bidirectional Reflectance Distribution Function (BRDF), Bidirectional Polarization Distribution Function (BPDF), black-sky, white-sky albedos and Normalized Difference Vegetation Index (NDVI). This research deserves to being published given the new valuable satellite products development, but there are still some descriptions and statements unclear and need to be improved. The detailed comments can be found below.

**Response:**

We would like to thank the reviewer for your time reviewing the manuscript and appreciate the constructive comments on our paper.

1. Introduction: I think most parts of the first paragraph needs to be re-written. POSP is a single-viewing multi-spectral polarimetric sensor aimed at aerosol detection, so I suppose this paragraph should introduce the research background about using multispectral or polarized measurements to retrieve aerosol properties. However, the authors mainly

discuss aerosol retrievals from multi-angle polarimetric measurements, while multi-angle is not one of the characteristics of POSP. I suggest to focus on the characteristics of POSP and introduce the underlying fundamental physics here.

**Response:**

Thanks for the suggestion! We have added some description about the POSP background and its main goal in the introduction. Generally, POSP was designed to enhance the atmospheric aerosol detection capability particularly for aerosol layer height, fine/coarse mode, to achieve the main goal of the GF-5(02) mission that is dedicated for $PM_{2.5}$ remote sensing (Li et al., 2022).

"POSP was designed to complement DPC measurement, which is a multi-angular polarimeter on the same platform with maximum 17 viewing angle and ~3.3 km spatial resolution measuring I, Q, U at VIS-NIR channels. POSP could extend to UV and SWIR channels, which is expected to enhance the atmospheric aerosol detection capability, particularly for aerosol layer height, fine/coarse mode, to achieve the main goal of the GF-5(02) mission that is dedicated for $PM_{2.5}$ remote sensing (Li et al., 2022). Meanwhile due to the on-board calibration device for POSP, it was expected to obtain higher accuracy of intensity/polarization measurements than DPC, which could perform cross-calibration between DPC and POSP (Lei et al., 2023). Therefore, the cross-track pattern was chosen to achieve more overlaps. On the other hand, there is a possibility to request to change POSP's scanning direction from cross-track to along-track.".

Lei, X., Liu, Z., Tao, F., Dong, H., Hou, W., Xiang, G., Qie, L., Meng, B., Li, C., Chen, F., Xie, Y., Zhang, M., Fan, L., Cheng, L., and Hong, J.: Data Comparison and Cross-Calibration between Level 1 Products of DPC and POSP Onboard the Chinese GaoFen-5(02) Satellite, Remote Sensing 2023, Vol. 15, Page 1933, 15, 1933, https://doi.org/10.3390/RS15071933, 2023.

Li, Z., Hou, W., Hong, J., Fan, C., Wei, Y., Liu, Z., Lei, X., Qiao, Y., Hasekamp, O. P.,
    Fu, G., Wang, J., Dubovik, O., Qie, L. L., Zhang, Y., Xu, H., Xie, Y., Song, M.,
    Zou, P., Luo, D., Wang, Y., and Tu, B.: The polarization crossfire (PCF) sensor
    suite focusing on satellite remote sensing of fine particulate matter PM2.5 from
    space, J Quant Spectrosc Radiat Transf, 286, 108217,
    https://doi.org/10.1016/J.JQSRT.2022.108217, 2022.

2. Section 2.1: It is mentioned that "POSP is the first space-borne multi-spectral crosstrack scanning polarimeter". Why does POSP use cross-track scanning method instead of along-track scanning? What are the advantages and disadvantages of cross-track scanning? Some background can be added here.

**Response:**

Thanks! Similar to previous comment, we have added some POSP background and its main goal in the introduction. Specifically, POSP is a cross-track scanning polarimeter flying on the GF-5(02) satellite for the first time. POSP was designed to complement DPC measurement, which is a multi-angular polarimeter on the same platform with maximum 17 viewing angle and ~3.3 km spatial resolution measuring I, Q, U at VIS-NIR channels. POSP could extend to UV and SWIR channels, which is expected to enhance the atmospheric aerosol detection capability, particularly for aerosol layer height, fine/coarse mode, to achieve the main goal of the mission that is dedicated for $PM_{2.5}$ remote sensing (Li et al., 2022). Meanwhile due to the on-board calibration device for POSP, it was expected to obtain higher accuracy of intensity/polarization measurements than DPC, which could perform cross-calibration between DPC and POSP (Lei et al., 2023). Therefore, the cross-track pattern was chosen to achieve more overlaps. On the other hand, there is a possibility to request to change POSP's scanning direction from cross-track to along-track.

3. Table 1: What is POSP/GF5 overpass time? I suggest to add it in this table and compare it with NOAA-20 since the difference of their overpass time can cause inconsistency in aerosol retrievals as mentioned in the analysis later.

**Response:**

Thanks! GF-5(02) satellite is in the descending node during the daytime and equatorial crossing time is 10:30 local time. We have added this information to Table 1.

4. Line 203-221: Although these land/ocean surface models including BRDF and BPDF have been discussed in many previous studies, I think some statements are still needed to clarify the meaning of related parameters involved in the state vector, such as aiso, avol and ageom (I assume they are the linear coefficients of three kernels in Ross-Li BRDF model), as well as r0, δFr and σ2. Is BPDF only considered for land surface but not ocean?

**Response:**

Yes, they are the linear coefficients of three Ross-Li BRDF kernels and BPDF is only considered over land. It's included in the main text in Section 3.1. We have included the descriptions below the list of state vectors. Thanks for the suggestion!

5. Figure 4: It seems the AE and scale height retrieval are less than other parameters, shown as many blank pixels over ocean in (e) and (f). What are the reasons for this situation? Are there any different criteria applied for different parameters retrieval availability or quality?

**Response:**

For extended aerosol parameters (SSA, AE and Scale Height), we use only POSP AOD (550 nm)>0.2 on daily basis to select high quality retrievals and then obtain the mean

values. Because the retrieval accuracy of these detailed parameters is strongly depending on the aerosol information content, simplistically aerosol loading/AOD. Based on our previous validation activities, satellite AOD higher than 0.2 is one of reasonable criteria to have a direct selection for SSA, AE and Scale Height. We have added this information to the Figure captions.

6. In the scatter plots, such as Figure 6-10, what does the color of each dot mean? This should be added in the caption.

Response:

Thanks for the suggestion! We have revised scatter validation to add heat map plots with the color represents the number of valid points in each 0.01x0.01 grid. Meanwhile, we added the validation plot in logarithmic scale for AOD, AODF and AODC.

---

## Author Response (AR2)

We would like to thank the referee and the editor for their time reviewing the manuscript, and for the helpful feedback provided. The detailed responses to all referees are provided below.

**Reviewer #2:**

I was a reviewer of the previous version of the manuscript. In this revision, the authors have addressed most of my previous issues. I appreciate these efforts. This includes improvements to the data file format – global processing attribute metadata have been added (some other dataset-level metadata are still missing such as variable long_names, although these are documented in the paper itself, so while not ideal I think it's acceptable if it is difficult for the authors to add these without a larger-scale processing effort).

As a result, I feel the paper is acceptable for publication in ESSD pending a few minor corrections, unless other reviewers identify problems I missed. As ESSD is primarily a data set description journal I see the requirements as being a little different from e.g. AMT/ACP and think the overall level of detail is appropriate here.

I think the below could probably be handled without further peer-review but would be happy to review again if the Editor would like.

**Response:**

We would like to thank the reviewer for your time reviewing the manuscript and appreciate the constructive comments on our paper.

1. Line 17: "continues" should be "continuous"

**Response:**

Revised.

2. Line 30: This sentence is too strong, it's not right to conclude that these products are all reliable. Only AOD, Angstrom, and fine/coarse split are validated against AERONET. The other things are compared against other satellites (not a validation!) or not at all (e.g. BPDF). I would just start this sentence as "Moreover, the data set provides not just total aerosol optical depth (AOD), but …" That is a more honest representation of what is shown in the paper.

**Response:**

Agree. This sentence has been revised to "Moreover, the developed POSP product includes not only total Aerosol Optical Depth (AOD), but also detailed properties of aerosol such as aerosol size, absorption, layer height, type, etc., as well as full surface Bidirectional Reflectance Distribution Function (BRDF), Bidirectional Polarization Distribution Function (BPDF), and black-sky, white-sky albedos and Normalized Difference Vegetation Index (NDVI).".

3. Line 61: "China has traditionally prioritized the development of Earth observation technologies and it has recently launched… " This statement veers into propaganda, not science (and is unsupported in any case). I suggest just starting "China has recently launched …"

**Response:**

Thanks! This sentence has been revised to "China has recently launched several payloads with polarimetric capabilities.".

4. Line 64: "past" is written twice, one can be removed.

**Response:**

Revised.

5. Table 1: is "spectral resolution" the full width at half maximum or something else? This should be defined more clearly as there are several possible interpretations. is it correct that the bandwidth for the 1380 nm channel is 40 nm? That seems quite wide compared to historical instruments, given the water vapor absorption here is quite narrow and the usual goal for this wavelength is to get a narrow band to increase sensitivity to only high-altitude features (e.g. cirrus).

**Response:**

Thanks! It should be full width half maximum (FWHW), we have added it to the Table 1. For the polarimetric band, due to the relative weak energy, the bandwidth is generally wider than non-polarimetric band. For example, the FWHW of 1380 nm for 3MI is 40 nm, while it's 20 nm for VIIRS.

6. Line 160: from reading the response to reviewer comments, it's not clear to me if (for the Dubovik AERONET inversion product) only the almucantar scans are used or also the hybrid scans. The hybrid scans were implemented to increase retrieval quality and availability during the day and are in the same file format (but a different data stream) from the almucantar scans. I initially assumed both were used but from reading the response to reviewers, this is no longer clear to me.

**Response:**

In this study, we use the inversion product from the almucantar measurements. We have added it to the main text.

7. Line 168: please add the Deep Blue data version used.

**Response:**

Thanks! We use NOAA20 VIIRS Deep Blue (DB) 6 km Version 2.0 Level 2 aerosol product (AERDB_L2_VIIRS_NOAA20) (Lee et al., 2024).

Lee, J., Hsu, N. C., Kim, W. V., Sayer, A. M., and Tsay, S. C.: VIIRS Version 2 Deep Blue Aerosol Products, Journal of Geophysical Research: Atmospheres, 129, e2023JD040082, https://doi.org/10.1029/2023JD040082, 2024.

8. Line 241: I would delete the word "new" as this technique has been used for 15+ years now by GRASP. So this is not a novel aspect to this specific application of GRASP.

**Response:**

We removed the word "new". Thanks!

9. Figure 16 and discussion: I'm not sure if the difference in the 2 micron band albedo is really from aerosols. At this wavelength, the aerosol contribution is small unless the coarse optical depth is high. I think that the spectral difference between 2130 (MODIS) and 2250 (POSP) nm band centers is probably the biggest factor (of course as the authors point out radiometric calibration can be important as well). This spectral difference can be quite significant because the absorption of different surfaces (and water) can vary a lot over this range. This can be seen in e.g. hyperspectral libraries (for example this soil spectrum from the ECOSTRESS library):

https://speclib.jpl.nasa.gov/ecospeclibdata/soil.aridisol.haplargid.none.all.89p1793.jhu.becknic.spectrum.txt and also in things like PACE OCI surface reflectance data which has SWIR bands at both these wavelengths (for example Figure 1 here):

https://www.tandfonline.com/doi/full/10.1080/2150704X.2025.2470905#d1e924

**Response:**

Thanks! We have added this discussion in the main text.